# A simplified approach for measuring Rubisco carbon isotope fractionation and the first determination in marine haptophyte *Gephyrocapsa oceanica*

Reto S. Wijker<sup>1,\*</sup>, Pere Aguiló-Nicolau<sup>2,\*</sup>, Madalina Jaggi<sup>1</sup>, Jeroni Galmés<sup>2</sup>, and Heather M. Stoll<sup>1</sup>

Correspondence: Reto S. Wijker (reto.wijker@eaps.ethz.ch) and Heather M. Stoll (heather.stoll@eaps.ethz.ch)

Abstract. Rubisco is the central photosynthetic enzyme that catalyzes the fixation of CO2 to RuBP, initiating the most dominant carbon assimilation pathway on Earth that supports nearly all trophic chains in the biosphere. The CO<sub>2</sub> fixation reaction expresses a strong kinetic isotope effect, producing biomass depleted in <sup>13</sup>C and leaving characteristic imprints in sediments and sedimentary rocks, which are widely used to reconstruct past biological activity and environmental conditions, including ancient atmospheric  $CO_2$  levels. Despite its importance, carbon isotope fractionation of Rubisco ( $\epsilon_{Rubisco}$ ) has been measured in only a limited number of organisms, with most studies focusing on land plants rather than on major contributors to the sedimentary record, such as cyanobacteria and coccolithophores. This scarcity reflects the complexity of existing experimental procedures and the high cost of instrumentation. Here, we present a simplified method that overcomes these limitations, eliminating the need for complex purification protocols, specialized equipment, and experimental designs that yield little CO2 fixation and high uncertainties. Using this protocol, we accurately determined  $\epsilon_{\text{Rubisco}}$  for the model plant *Spinacia oleracea*, the cyanobacterium Synechococcus sp., and provide the first determination for the coccolithophore Gephyrocapsa oceanica. The measured values span a striking range, from 13.1 % to 30 %, highlighting both the variability of Rubisco fractionation and the versatility of our approach for studying carbon isotope discrimination across diverse biological systems. This study establishes a method that enables reliable determination of  $\epsilon_{\rm Rubisco}$  across phylogenetically diverse groups, thereby supports research that provides new insights into the mechanisms of Rubisco fractionation, and improves interpretation of environmental carbon isotope records.

## 1 Introduction

Ribulose-1,5-bisphosphate carboxylase oxygenase (Rubisco) is the key photosynthetic enzyme that catalyses the addition of CO<sub>2</sub> to ribulose-1,5-bisphosphate (RuBP), producing two molecules of 3-phosphoglycerate (3-PGA). This reaction underpins the primary step of carbon fixation and supports almost all trophic chains in the biosphere (Prywes et al., 2023). While Rubisco also participates in other biochemical pathways, its most critical role is oxygenic photosynthesis via the Calvin-

<sup>&</sup>lt;sup>1</sup>Department of Earth and Planetary Sciences, ETH Zürich, Sonnegstrasse 5, 8092, Zürich, Switzerland

<sup>&</sup>lt;sup>2</sup>Research Group on Plant Biology under Mediterranean Conditions, Universitat de les Illes Balears–INAGEA, Palma, Balearic Islands, Spain

<sup>\*</sup>These authors contributed equally to this work.

Benson-Bassham (CBB) cycle (Berg, 2011), contributing to the fixation of  $\sim$ 220 Gt of carbon annually (Bar-On and Milo, 2019).

In addition to CO<sub>2</sub> fixation, Rubisco also catalyses the oxygenation of RuBP, producing one molecule of 3-PGA and one of 2-phosphoglycolate (2-PG). The latter requires detoxification via the photorespiratory pathway, which results in energy loss and the release of previously fixed CO<sub>2</sub> (Bauwe et al., 2012). Beyond its dual substrate specificity, Rubisco is characterized by a low carboxylation turnover rate ( $k_{cat}^c$ ) and low affinity for CO<sub>2</sub> (inverse of Michaelis-Menten constant for CO<sub>2</sub>; K<sub>C</sub>). For these reasons, it is often described as an inefficient enzyme, although this designation remains a topic of debate (Bathellier et al., 2018).

30 Another feature of Rubisco's carboxylation activity is its kinetic isotope effect (KIE), whereby it preferentially fixes the lighter carbon isotope, <sup>12</sup>CO<sub>2</sub>, over <sup>13</sup>CO<sub>2</sub> (Farquhar et al., 1989). This discrimination results in photosynthetic biomass that is significantly depleted in <sup>13</sup>C. The carbon isotope fractionation of Rubisco (ε<sub>Rubisco</sub>) arises from the higher zero-point vibrational energy of <sup>12</sup>CO<sub>2</sub>, which lowers the activation energy for the transition state and facilitates bond formation. The KIE is further amplified when the transition state closely resembles the carboxylation product, thereby stabilizing the intermediate (Tcherkez and Farquhar, 2005).

Rubisco's isotope effect has been preserved in the carbon isotope signatures of sedimentary rocks, allowing the reconstruction of biological and environmental conditions as far back as 3.8 billion years ago — shortly after Earth formed approximately 4.5 billion years ago (Wang et al., 2023b). Despite its biogeological importance,  $\epsilon_{\text{Rubisco}}$  has only been measured in a limited number of organisms, with most studies focused on land plants rather than in the major contributors to sedimentary records: cyanobacteria for Precambrian era, and coccolithophores and diatoms as marine primary producers during the Phanerozoic (Garcia et al., 2021). However, recent work is being done to explore  $\epsilon_{\text{Rubisco}}$  in such phylogenetic groups (Aguiló-Nicolau et al., in preparation).

Reported  $\epsilon_{\text{Rubisco}}$  values in extant enzymes range from 11 to 30 % (Boller et al., 2015, 2011; Thomas et al., 2019; Roeske and O'Leary, 1984; Wang et al., 2023b; Scott et al., 2004b; Guy et al., 1993). This wide variation has been documented in only a limited number of species, suggesting that additional values are yet to be discovered. However, because  $\epsilon_{\text{Rubisco}}$  is governed by a hyperbolic-sine relationship typical of mass-dependent fractionation, its potential range of variation is more constrained than that of Rubisco kinetic parameters, which follow an Arrhenius-type function (Galmés et al., 2016; Tcherkez and Farquhar, 2005). One of the main reasons for the limited number of reported  $\epsilon_{\text{Rubisco}}$  values is the complexity of the experimental procedures involved and the high cost of the required instrumentation.

Early determinations of carbon isotope fractionation by Rubisco date back to the 1960. Park and Epstein (1960) isolated tomato Rubisco and incubated it with bicarbonate and RuBP at 25 °C for 1 h. Assuming complete conversion of RuBP to 3-PGA, they acidified the mixture to release unreacted bicarbonate, isolated and purified the 3-PGA, combusted it, and analysed the resulting CO<sub>2</sub> using a Nier-type mass spectrometer. The same method was later applied to other species such as *Sorghum bicolor*, *Glycine max* and *Gossypium* sp. (Christeller et al., 1976; Whelan et al., 1973; Wong et al., 1979). A major refinement came with the work of Roeske and O'Leary (1984, 1985) who conducted similar incubations using pure spinach and *Rhodospirillum rubrum* Rubisco for up to 9 h at 25 °C. After removing protein, they precipitated 3-PGA using barium

salt, followed by enzymatic decarboxylation to release the fixed CO<sub>2</sub> from 3-PGA. This CO<sub>2</sub> was collected and analysed using isotope ratio mass spectrometry (IRMS). To account for 3-PGA formed via non-carboxylation pathways, they also ran a parallel oxygenase-only reaction. Later, Guy et al. (1993) introduced a method based on tracking changes in the isotope composition of the substrate during its consumption — a method known as substrate depletion, based on Rayleigh fractionation principle. In this approach, purified Rubisco was incubated in sealed vessels, and two to five samples were taken over the course of the reaction. Each was acidified and transferred to a dry-ice-ethanol trap to collect CO<sub>2</sub> before being analysed on an IRMS. This method has since become widely adopted, with various improvements and adaptations (McNevin et al., 2006; Scott et al., 2004b; Thomas et al., 2019; Wang et al., 2023a, b). von Caemmerer et al. (2014) introduced the simultaneous measurement of gas exchange and isotope discrimination using a tuneable diode laser coupled to a Li-6400 system, applying the data to a ternary-corrected discrimination model (Farquhar and Cernusak, 2012).

Despite these developments, all existing methods for determining  $\epsilon_{\rm Rubisco}$  present limitations, which this study aims to address. Early approaches lacked key methodological details and relied on several problematic assumptions: complete conversion of RuBP to 3-PGA, no consideration of oxygenation activity or enzyme deactivation over time, and omission of carbonic anhydrase to facilitate rapid interconversion of dissolved inorganic carbon (DIC) to  ${\rm CO_2}$ —the true substrate of Rubisco (O'Leary, 1981). Although the substrate depletion method no longer requires accounting for oxygenation-derived 3-PGA, some applications have reported high variability in  $\epsilon_{\rm Rubisco}$  estimates within the same species (Boller et al., 2011, 2015). In cases where RuBP was limiting, the assumption of full substrate consumption becomes questionable, particularly given the lack of correction for inhibitor formation by Rubisco side reactions over time (Wang et al., 2023a, b; Pearce, 2006). Moreover, several studies report experiments with less than 30 % DIC consumption — and in some cases even less than 6 % — casting doubt on the reliability of the linearization required for Rayleigh fractionation, increasing uncertainty and potentially compromising the accuracy of the derived  $\epsilon_{\rm Rubisco}$  values.

The need for highly purified enzymes adds complexity and time to an already demanding protocol, and its necessity has not been experimentally validated yet. Furthermore, the use of IRMS entails high equipment maintenance costs.

Despite the significance of  $\epsilon_{\text{Rubisco}}$  for biogeochemical and evolutionary models, no standardized, accessible protocol exists across diverse phylogenetic groups. Here we propose a method that provides several advantages over existing approaches and enables  $\epsilon_{\text{Rubisco}}$  determination across a wide range of taxa, using a single instrument for simultaneous quantification of total DIC and its isotopic composition and with sample concentration and sampling times supported by a realistic kinetic model of DIC consumption by Rubisco from different taxa. We illustrate the reproducibility of the method for *S. oleracea* and *Synechococcus* sp. and provide a first determination for coccolithophore *Gephyrocapsa oceanica*.

#### 2 Materials and methods

## 2.1 Strains, Media, and Growth Conditions

Three to four *S. oleracea* (cultivar Winter Giant Santos) seeds were sown in each of 45 individual 2-L pots containing a universal growing medium and maintained under automated watering. Plants were grown in a temperature- and humidity-controlled

chamber (Fitoclima 10,000 HP, Aralab, Spain) under 500  $\mu$ mol photons m<sup>-2</sup>s<sup>-1</sup> of photosynthetically active radiation, at 25 °C and 60% relative humidity for one month. Fully illuminated, non-senescent leaves were harvested.

*G. oceanica* (RCC 1303) purchased from the Roscoff Culture Collection, was grown in batch cultures in sterile flasks placed on a roller at 10 rpm to ensure uniform mixing and light exposure. Cultures were maintained in K/2 medium as described in Keller et al. (1987). Artificial seawater (pH 8.1) was used instead of natural seawater, following the composition described by Kester et al. (1967). Cultures were incubated under LED light strips programmed with a sinusoidal 14-hour light / 10-hour dark cycle, reaching a maximum light intensity of 120  $\mu$ mol photons m<sup>-2</sup>s<sup>-1</sup>. Growth temperature was set to 21 °C. Cell growth was monitored using a Z2 Coulter particle counter (Beckman Coulter, Inc., Brea, California, United States). Cultures were maintained semi-continuously by harvesting 80% – 90% of the culture volume and refreshing it when cell density reached

Synechococcus sp. PCC 6301 was obtained from Pasteur Culture Collection of Cyanobacteria (PCC, France) and grown under constant agitation in 50 or 250 mL graduated and ventilated cell culture flasks (CCFP-25V-100, Labbox, Spain). The strain was cultured in Z-medium (Staub, 1961) under a 16-hour light / 8-hour dark cycle, with a light intensity of 50 μmol m<sup>-2</sup> photons s<sup>-1</sup> (4000 K, Osram L 18W/840 Lumilux, Germany) and a constant temperature of 24 °C (Aralab Fitoclima S600, PLH, Spain). Growth was monitored continuously by measuring culture absorbance at 650 nm (OD<sub>650</sub>, Multiskan Sky 1530-105 00433C, Thermo Scientific, USA). Once the OD<sub>650</sub> reached 0.5, biomass was collected by centrifugation at 8,000 × g for 3 minutes at 24 °C. All harvested biomass was snap-frozen in liquid nitrogen, and stored at -80 °C until Rubisco extraction.

approximately 450,000 cells per mL. Cells were harvested by vacuum filtration through 2  $\mu$ m mesh filters.

## 2.2 Extraction and purification

## 2.2.1 Semi-purification

Rubisco from *S. oleracea* was extracted following the protocol described by Capó-Bauçà et al. (2023). Briefly, 0.3 g of fresh leafs were snap-frozen in liquid N<sub>2</sub> and ground into a fine powder using a mortar and pestle. The powder was then mixed with 0.2 g of polyvinylpolypyrrolidone (PVPP) and an equal amount of pre-washed sand. The mixture was lysed with 2 mL of ice-cold extraction buffer containing 100 mM EPPS (pH 8.1), 15 mM MgCl<sub>2</sub>, 1 mM ethylenediaminetetraacetic acid (EDTA), 10 mM dithiothreitol (DTT), 100 mM β-mercaptoethanol, 2% protease inhibitor cocktail (PIC, P9599, Merck, USA), 4 mM phenylmethylsulfonyl fluoride (PMSF), and 0.5% Triton X-100. The lysate was centrifuged at 15,000 x g for 3 minutes at 4 °C, and the resulting supernatant was aliquoted.

Rubisco from *G. oceanica* was extracted by resuspending 10 filters containing cells in 4 mL of ice-cold extraction buffer using a vortex mixer. Empty filters were then removed, and approximately 1 mL of 2 mm glass beads was added. Cell lysis was performed on ice using a probe sonicator (UP200St, Hielscher Ultrasonics, Germany) for 8 minutes, with alternating 30-second on/off intervals at 40% of the maximum power output. The lysate was centrifuged at 8,000 x g for 20 minutes at 4 °C. The resulting supernatant was aliquoted.

Rubisco from *Synechococcus* sp. was extracted using the same procedure, but with a modified extraction buffer composed of 100 mM Bicine pH 8.1, 20 mM MgCl<sub>2</sub>, 1 mM EDTA, 1 mM Benzamidine, 1 mM  $\epsilon$ -aminocaproic acid, 10 mM DTT, 2 % PIC,

125

130

100 mM  $\beta$ -mercaptoethanol, 20 mM PMSF, 2 % CelLyticTM B (B7435, Merck, USA), 2.5 mL of 2 mm glass beads, and 0.1 g PVPP as described in Aguiló-Nicolau et al. (in preparation). All aliquoted supernatant were snap-frozen in liquid nitrogen and stored at -80 °C

Rubisco from *S. oleracea* and *G. oceanica* was partially-purified via anion-exchange chromatography using Bio-Scale Mini Macro-Prep High Q cartridges (7324124, Bio-Rad, USA). The protein was then desalted and concentrated approximately 10-fold by centrifugation at 2,000 x g and 4 °C using 10 kDA Amicon Ultra-4 centrifugal filter units (UFC8010, Merck, USA). For *Synechococcus* sp., the same anion-exchange procedure was employed, with specific variations as described in Aguiló-Nicolau et al. (in preparation)

## 2.2.2 Full-purification

The full-purification protocol from Amaral et al. (2024) was followed and adapted to fully purify Rubisco from S. oleracea leaves. Approximately 20-30 g of fresh leaf tissue were placed in a 100 mL beaker together with 100 mL of ice-cold extraction buffer. The mixture was homogenized on ice using a blender. The homogenate was filtered through several layers of wet Miracloth (475855, Merck, USA) into a pre-chilled 50 mL beaker, transferred to centrifuge tubes, and centrifuged at 18,000 x g for 20 minutes at 4 °C. The resulting supernatant was transferred into a cold graduated cylinder, and 60% polyethylene glycol (PEG) was added at a volume equal to 50% of the supernatant. 1 M MgCl<sub>2</sub> was then added to a final concentration of 20 mM followed by gentle mixing by inversion and incubated under magnetic stirring at 4 °C for 30 minutes. After incubation, the mixture was centrifuged at 18,000 × g for 30 minutes at 4 °C with slow deceleration, and the resulting pellet was resuspended in 20 mL of buffer (100 mM EPPS pH 8.1, 15 mM MgCl<sub>2</sub>, and 1 mM EDTA) containing 2% PIC using a pre-chilled tissue homogenizer. This homogenate was subjected to ultracentrifugation at 200,000 x g for 20 minutes at 4 °C with slow deceleration. The final supernatant was loaded onto two 5 mL Bio-Scale Mini Macro-Prep High Q anion-exchange cartridges (7324124, Bio-Rad, USA) connected in series to an ÄKTA pureTM 25 FPLC system (29018226, Cytiva, USA). Protein elution was monitored at 280 nm (OD<sub>280</sub>), and Rubisco-containing fractions were collected and further desalted using a SuperdexTM 200 Increase 10/300 GL column (28990944, Cytiva, Sweden). The desalted fractions were concentrated approximately 10-fold by centrifugation at  $2,000 \times g$  and 4 °C using 10 kDa Amicon Ultra-4 filters, snap-frozen in liquid nitrogen, and stored at -80°C.

# 2.3 Isotope fractionation experiment

The KIE of *S. oleracea*, *G. oceanica*, and *Synechococcus* sp. Rubisco was determined based on the substrate depletion method described by Scott et al. (2004b). This method tracks changes in  $\delta^{13}$ C of dissolved inorganic carbon (DIC) during its progressive consumption by the Rubisco-catalyzed carboxylation reaction. Assays were conducted in 10 mL gas-tight syringes (Hamilton, USA) to prevent contamination by atmospheric  $CO_2$ . The reaction buffer consisted of 100 mM EPPS at pH 7.8, 20 mM MgCl<sub>2</sub>, and approximately 6 mM NaHCO<sub>3</sub>. Buffers were purged with N<sub>2</sub> gas overnight before use to eliminate  $CO_2$ . Carbonic anhydrase (40  $\mu$ g/mL, from bovine erythrocytes; C3934, Sigma-Aldrich) was included to ensure rapid chemical and isotopic equilibration between  $CO_2$  and  $HCO_3^-$ . Partially or fully-purified Rubisco, pre-activated by incubation with 20 mM NaHCO<sub>3</sub>

for 30 minutes at room temperature, was added to the reaction syringe at a final concentration of 70–80  $\mu$ g/ml of active enzyme. The reaction was initiated by injecting an equimolar concentration of RuBP synthesized and purified according to Kane et al. (1994) relative to the DIC. At defined time intervals, samples were withdrawn to monitor both DIC concentration and  $\delta^{13}$ C. Each sample was divided into two fractions. The first fraction, containing at least 1.25  $\mu$ mol DIC (0.5-2 mL), was diluted into 2 mL of N<sub>2</sub>-purged 110 mM EPPS buffer (pH 7.8) and either injected directly into a DIC- $\delta^{13}$ C Analyzer (AS-D1 and G2131-i Apollo-Picarro, USA) or filtered through a 50 kDa Amicon Ultra-4 centrifugal unit (UFC8010, Merck, USA) at 2,000 × g for 3 minutes at 25 °C before injection for concentration and carbon isotope analysis of DIC. The second fraction, containing at least 0.3  $\mu$ mol DIC (0.1-0.5 mL), was immediately injected into a 5 mL septum-capped vial flushed with helium and preloaded with 0.1 mL of 200 mM H<sub>3</sub>PO<sub>4</sub>. These samples were analyzed the following day using a GasBench II system coupled via a ConFlow IV interface to a Delta V Plus isotope ratio mass spectrometer (Thermo Fisher Scientific, USA). Reactions were carried out in a temperature-controlled incubator or climate chamber set precisely to 25 °C. Syringes were continuously rotated at 20 rpm on a roller to ensure homogeneous mixing. Control assays were performed under identical conditions but without RuBP or Rubisco extract. All experiments were conducted in duplicate or triplicate.

## 2.4 Analytical methods

170

# 2.4.1 Total soluble protein and active Rubisco quantification

As described in detail by Aguiló-Nicolau et al. (in preparation), the concentration of Rubisco active sites was determined by incubating extracts with 25 mM NaHCO<sub>3</sub> for 30 minutes at 25 °C, followed by a 30-minute incubation with the Rubisco specific binding inhibitor <sup>14</sup>C-radiolabelled 2'-carboxy-D-arabinitol-1,5-bisphosphate (<sup>14</sup>C-CABP) (Ruuska et al., 1998). The unbound inhibitor was then separated from Rubisco-bound <sup>14</sup>C-CABP by column chromatography using Sephadex G-50 Fine (17-0042-01, GE Healthcare, USA). The radioactivity of the fraction containing Rubisco-bound inhibitor was quantified using a scintillation counter (Tri-Carb 4910 TR, Revity, USA). Total soluble protein (TSP) content in the extracts was determined using the Bradford assay (Bradford, 1976).

## 2.4.2 Concentration and stable isotope ratio measurement

Concentration and  $\delta^{13}$ C composition of DIC were measured using an Apollo acidification system AS-D1 (Apollo SciTech, LLC, USA) coupled to a Picarro G2131-i cavity ring-down spectrometer (Picarro Inc., USA). For each measurement, 0.5 to 2 mL of sample (containing at least 1.25  $\mu$ mol DIC) was diluted in 2 mL of 110 mM EPPS buffer and transferred to an acidification chamber, where 0.9 mL of 5 M phosphoric acid was added to convert DIC into CO<sub>2</sub> gas. The evolved CO<sub>2</sub> was then sparged and transferred to the Picarro analyzer for isotopic and concentration analysis.

Two concentrations of in-house NaHCO<sub>3</sub> isotope standards, prepared in both deionized water and 110 mM EPPS buffer, were analyzed at the beginning and end of each run to monitor instrument accuracy and correct for any drift. The average standard deviation of these standards was 0.2‰. Additionally, a certified DIC reference material from the Scripps Institution of Oceanography (Dickson, 2010) was injected at multiple concentrations to calibrate the quantitative DIC measurements. The

EPPS buffer used to dilute the samples was analyzed repeatedly as a blank. Although it was degassed by bubbling with  $N_2$  overnight, trace amounts of DIC were still detectable and were subtracted from measured sample concentrations.

To validate the Apollo-Picarro results, the carbon isotope composition of DIC was also measured using a GasBench II system (Thermo Fisher Scientific, Germany) equipped with an autosampler (CTC Analytics AG, Switzerland), coupled to a ConFlo IV interface and a Delta V Plus isotope ratio mass spectrometer (Thermo Fisher Scientific). The same in-house NaHCO<sub>3</sub> isotope standards used in the Apollo-Picarro setup were also employed in a standard bracketing procedure. Average offsets between known and measured  $\delta^{13}$ C values of the standards were used to correct all sample measurements. The average standard deviation of these standards was 0.2%. All carbon isotope signatures are reported in per mil (‰) relative to Vienna Pee Dee Belemnite (VPDB) ( $\delta^{13}$ C<sub>VPDB</sub>).

#### 2.5 Data evaluation

The calculation of  $\epsilon_{\rm Rubisco}$  is based on the Rayleigh distillation effect, which describes the progressive change in the isotopic composition of  ${\rm CO_2}$  as it is consumed during the reaction. The  $\epsilon_{\rm Rubisco}$  value was determined by linear regression analysis of the natural logarithm of the carbon isotope ratios of  ${\rm CO_2}$  versus the natural logarithm of the remaining  ${\rm CO_2}$  concentration, following Scott et al. (2004a) and Equation 1:

$$\ln(R_{CO_2}) = \left(\frac{1}{\alpha} - 1\right) \cdot \ln\left[CO_2\right] + \ln\left(\frac{R_{CO_2,0}}{\left[CO_2\right]_0^{1/\alpha - 1}}\right) \tag{1}$$

where  $R_{\rm CO_2}$  and  $[{\rm CO_2}]$  are the carbon isotope ratio and concentration of  ${\rm CO_2}$  at time t, and  $R_{\rm CO_2,0}$  and  $[{\rm CO_2}]_0$  are the corresponding initial values. Since  ${\rm CO_2}$  is the actual substrate for Rubisco, but our measurements were performed on DIC, it was necessary to correct for the equilibrium isotope fractionation between  ${\rm CO_2}$  and  ${\rm HCO_3^-}$ , as described in Scott et al. (2004b), using the modified Rayleigh equation for DIC (Equation 2):

$$\ln(R_{DIC}) = \left(\frac{1}{\alpha \cdot C_{i}} - 1\right) \cdot \ln\left[DIC\right] + \ln\left(\frac{R_{DIC,0}}{\left[DIC\right]_{0}^{1/\alpha - 1}}\right)$$
(2)

where  $C_i$  is the equilibrium isotope effect between dissolved  $CO_2$  and  $HCO_3^-$  at reaction temperature i, as reported by Guy et al. (1993) and Mook et al. (1974).  $R_{DIC}$  and [DIC] are the isotope ratio and concentration of DIC at time t, and  $R_{DIC,0}$  and [DIC]<sub>0</sub> are their corresponding initial values. The slope m obtained from the linear regression of  $ln(R_{DIC})$  versus ln([DIC]) was used to calculate  $\alpha$  and  $\epsilon_{Rubisco}$  according to equations 3 and 4:

$$\alpha = \frac{1}{C_i(m+1)} \tag{3}$$

$$\epsilon_{\text{Rubisco}} = (\alpha - 1) \cdot 1000\%$$
 (4)

(9)

Data from replicate experiments were combined using the Pitman estimator (Scott et al., 2004a). Uncertainties were calculated by Gaussian error propagation and are reported as standard deviation.

#### 2.6 Kinetic modeling of DIC depletion

To improve calculation of the needed Rubisco concentration and optimal sampling times, we modeled the concentration dynamics of DIC during the *in vitro* Rubisco catalyzed  $CO_2$  fixation assay using Michaelis-Menten kinetics, incorporating the formation of a inhibitory side product — xylulose 1,5-biphosphate (XBP) — known as a competitive inhibitor of Rubisco activity (Pearce, 2006). At each time point  $t_i$ , the concentrations of DIC and  $CO_2$  were recalculated based on their values at the preceding time point  $t_{i-1}$ . The initial DIC concentration at  $t_{i-1}$  was set to the measured value at the start of the assay. DIC depletion over time was modelled as:

$$[DIC]_{t_i} = [DIC]_{t_i-1} - v_{t_{i-1}}(t_i - t_{i-1})$$
(5)

The corresponding  $CO_2$  concentration at each time point was calculated from the DIC pool using the carbonate equilibrium, as follows:

$$[CO_2]_{t_i} = \frac{[DIC]_{t_i}}{1 + \frac{K_1}{[H^+]} + \frac{K_1 \cdot K_2}{[H^+]^2}}$$
(6)

where  $[H^+]$  was derived from the measured pH of the reaction mixture. The equilibrium constants  $K_1$  and  $K_2$  (Ric, 2001) were adjusted for temperature and ionic strength as described by Yokota and Kitaoka (1985). Reaction rates were modeled using Michaelis-Menten kinetics, with and without competitive inhibition:

220

$$\mathbf{v}_{t_i} = \frac{\mathbf{v}_{\text{max}} \cdot [\text{CO}_2]_{t_i}}{\mathbf{K}_{\text{M}} + [\text{CO}_2]_{t_i}} \tag{7}$$

$$\mathbf{v}_{t_i} = \frac{\mathbf{v}_{\text{max}} \cdot [\text{CO2}]_{t_i}}{\mathbf{K}_{\text{M}} \left( 1 + \frac{[\mathbf{I}]_{t_i}}{\mathbf{K}_{\mathbf{I}}} \right) + [\text{CO}_2]_{t_i}} \tag{8}$$

where  $K_M$  is the Michaelis-Menten constant, taken from Hermida-Carrera et al. (2016), Aguiló-Nicolau et al. (2023), and unpublished data (in preparation).  $K_I$  represents the inhibition constant for XBP, specific to the Rubisco form, and was taken from Pearce (2006).  $v_{max}$  is the maximum reaction rate. The concentration of the inhibitor at each time point was modeled as:

$$[I]_{t_i} = \mathbf{k}_{acc} \cdot (t_i - t_0)$$

where  $k_{acc}$  is the accumulation rate of inhibitor XBP. The parameters  $v_{max}$  and  $k_{acc}$  were estimated by fitting the model to experimental data. Parameter optimization was achieved by minimizing the root mean square deviation between the modeled and measured DIC concentrations.

#### 3 Results and discussion

## 240 3.1 Optimizing Rubisco purity for reliable isotope fractionation measurements

To establish a protocol for measuring  $\epsilon_{\text{Rubisco}}$ , we used enzyme extracted from *S. oleracea* leaves. This species was selected because of its high biomass yield and high Rubisco content, and because it enables direct comparison of our fractionation results with previously published values. To minimize potential interference in the *in vitro*  $\text{CO}_2$  fixation assays from other carboxylases that could alter the  $\delta^{13}\text{C}$  composition, as well as to reduce the influence of inhibitory proteins, Rubisco was purified from other proteins present in the crude leaf extract. Two different purification protocols were tested: (1) a fast and simple method that yields partially-purified Rubisco, and (2) a more time-intensive procedure that produces a highly purified Rubisco extract (see Methods for details).

To visually assess the extent of purification achieved by the two protocols, we performed SDS-PAGE. The gel, shown in Figure 1, was loaded with comparable amounts of non-purified crude cell extract (CE), semi-purified Rubisco extract (SPE), and fully- purified Rubisco extract (FPE) obtained from the two purification procedures. The fully-purified extract shows almost exclusively two bands at approximately 55 kDa and 14 kDa, corresponding to the large and small subunits of Rubisco, respectively. In contrast, the semi-purified lane displays, in addition to the two prominent Rubisco bands, several faint bands that indicate the presence of other proteins, although these are substantially less intense than the numerous additional bands observed in the crude extract. Overall, the gel demonstrates a clear increase in Rubisco purity from crude extract to semi-purified to fully-purified extract.

To quantify the degree of Rubisco purification in the three extracts, we compared the total protein content with the amount of Rubisco present. Total protein concentration was determined using the Bradford assay, and Rubisco content was quantified from SDS PAGE analysis. We estimated that Rubisco accounts for approximately  $34 \pm 9\%$  of the total protein in crude extract,  $70 \pm 8\%$  in the semi-purified extract, and  $91 \pm 9\%$  in the fully-purified exract. This represents a roughly twofold enrichment in the semi-purified extract and nearly threefold in the fully-purified extract relative to the crude extract. Based on the  $^{14}$ C-CABP binding assay, approximately 63% of the Rubisco in the fully-purified extract and nearly 100% in the semi-purified extract was catalytically active. In the following sections, we describe and discuss the results of *in vitro* isotope fractionation experiments conducted using both the fully-purified and semi-purified Rubisco extracts.

# 3.2 Optimization of Rubisco concentration and sampling using a kinetic model

The concentration of Rubisco used in the assay was a critical parameter, as it directly influenced the rate of CO<sub>2</sub> consumption, the extent of dissolved inorganic carbon (DIC) depletion, and the optimal timing of sample collection. To optimize both the enzyme concentration and the sampling schedule, we developed a kinetic model to simulate DIC dynamics throughout the course of the reaction.

Figure 2a shows the time course of DIC consumption during CO<sub>2</sub> fixation for two different concentrations of fully-purified Rubisco from *S. oleracea*. The symbols represent experimentally measured DIC concentrations sampled throughout the reaction, while the black lines indicate the modelled DIC decline based on standard Michaelis-Menten kinetics. In the early stages

**Figure 1.** Qualitative SDS-PAGE analysis of protein extracts from *S. oleracea* leaves purified for Rubisco using two different protocols: one for semi-purification and the other for full purification. Proteins were separated on an SDS-PAGE gel and stained with Coomassie Brilliant Blue. Lane 1: molecular weight marker; Lane 2: internal control (IC); Lanes 3–4: crude extract (CE); Lanes 5–6: semi-purified extract (SPE); Lanes 7–10: fully-purified extract (FPE). Prominent bands at approximately 55 kDa and 14 kDa correspond to the large and small subunits of Rubisco, respectively.

of the reaction, DIC is consumed rapidly and aligns well with the model predictions. As the reaction proceeds, however, the fixation rate progressively declines and eventually approaches zero, deviating substantially from the expected Michaelis-Menten behavior. This discrepancy is attributed to a progressive loss of Rubisco catalytic activity over time. It is well established that intermediate by-products formed during the enolization-carboxylation reaction — such as XBP — can act as inhibitors, leading to self-inhibition of Rubisco (Pearce, 2006). To account for this effect, we extended the Michaelis-Menten model by incorporating a competitive inhibition term, assuming a linear accumulation of this inhibitory by-product throughout the reaction. Although XBP accumulation is likely non-linear and substrate dependent, a linear approximation was used in the absence of more detailed kinetic constraints. The inhibitor accumulation rate ( $k_{\rm acc}$ ) was treated as a free parameter. Similarly, while  $v_{\rm max}$  could in principle be constrained from known  $k_{\rm cat}$  values and Rubisco concentrations, the resulting fits were unsatisfactory, so  $v_{\rm max}$  was also treated as a free fitting parameter.

The best-fit solution for the two free parameters was obtained by minimizing the root mean square (RMS) difference between the experimental and model-predicted DIC concentrations over the course of the reaction. The optimized parameters yielded an RMS deviation of approximately  $0.07 \pm 0.04$  mM across all experiments and a strong linear correlation between measured and modelled DIC values, with an  $R^2$  of 0.997 (see Figure 3). The modelled DIC depletion, shown as green lines in Figure 2a, captures the experimental observations significantly better than the standard Michaelis-Menten model without inhibition (black lines in Figure 2a). The best-fit parameters for  $v_{\rm max}$  and the production rates are summarized in Table 1. Based on the fitted  $v_{\rm max}$  values and the known Rubisco concentrations, we calculated an average  $k_{\rm cat}$  of  $2.3 \pm 0.5$  s<sup>-1</sup>, which is in excellent agreement with the reported  $k_{\rm cat}$  of  $2.4 \pm 0.1$  s<sup>-1</sup> for *S. oleracea* Rubisco (Hermida-Carrera et al., 2016).

Detailed knowledge of reaction kinetics is essential for determining both the optimal timing of sample collection and the appropriate amount of enzyme to include in the assay. In the high-activity reaction shown in Figure 2a (dark green color), approximately three times more Rubisco was used compared to the low-activity reaction (green color), resulting in approximately 90% DIC depletion before the reaction significantly slowed (Table 1). In contrast, the lower Rubisco concentration led to a

**Figure 2.** DIC depletion during the course of CO<sub>2</sub> fixation catalyzed by Rubisco from: (a) fully-purified *S. oleracea* extract, (b) semi-purified *S. oleracea* extract, (c) semi-purified *G. oceanica* extract, and (d) semi-purified *Synechococcus* sp. extract. Symbols represent measured DIC concentrations, lines represent modelled concentrations. Circles show DIC measurements from active reaction assays, violet diamonds represent control experiments without RuBP, and yellow triangles represent controls without Rubisco extract. Black lines correspond to fits using standard Michaelis-Menten kinetics, while the other lines incorporate competitive product inhibition by reaction by-products.

final DIC depletion of only 77%. However, the time required to process each sample — approximately 7-8 minutes — limited the number of samples that could be collected during the high-activity assay. This restriction hindered accurate calculation of isotope enrichment factors, as the limited number of data points produced suboptimal results. To obtain robust estimates of carbon isotope enrichment, we aimed to collect approximately 10 samples and reach at least 70% DIC depletion. This balance was successfully achieved in the low-activity assay shown in Figure 2a, which provided high-quality data, as detailed in the following section. Based on these constraints, an active Rubisco concentration of approximately 70-80  $\mu$ g/ml was found to be optimal. Significantly lower concentrations led to inadequate DIC fixation and unacceptably long reaction durations.

310

**Figure 3.** Comparison of predicted versus experimentally measured DIC concentrations across all reaction assays. Each point represents a measurement of DIC concentration during a Rubisco catalyzed CO<sub>2</sub> fixation reaction. The 1:1 diagonal line indicates perfect agreement between model predictions and observed values. Data include assays from *S. oleracea*, *G. oceanica*, and *Synechococcus* sp.

# 3.3 Comparison of Apollo-Picarro and GasBench Systems for isotope fractionation analysis

We simultaneously measured both the concentration and isotopic composition of DIC using the Apollo acidification system coupled to a Picarro cavity ring-down spectrometer (Apollo-Picarro system). To evaluate the accuracy of the isotope measurements from this setup, we also analyzed the isotope data using a more traditional method: the GasBench system coupled to a Delta V Plus isotope ratio mass spectrometer. Fully-purified Rubisco from *S. oleracea* was used for this comparison.

Depletion of DIC during  $CO_2$  fixation was accompanied by substantial  $^{13}C$  enrichment in the residual DIC pool, as shown in Figure 4a and b for the GasBench and Apollo-Picarro systems, respectively. The  $\delta^{13}C$  values increased by up to +58% at 77% substrate conversion. A strong linear correlation ( $R^2 > 0.99$ ) was observed in the regression analysis of the logarithm of carbon isotope ratio versus the logarithm of the remaining DIC concentration in both systems (Figure 4c and d). Using Equation 2–4, we calculated  $\epsilon_{\text{Rubisco}}$  values summarized in Table 1. The  $\epsilon_{\text{Rubisco}}$  determined using the Apollo-Picarro system for both isotope and concentration measurements was  $29.6 \pm 0.7\%$ , while a value of  $30.4 \pm 0.5\%$  was obtained when isotope ratios were measured with the GasBench system and concentrations with the Apollo-Picarro system. Although the latter value is slightly higher, the difference is not statistically significant.

Averaging the two systems yields an ε<sub>Rubisco</sub> of 30.0±0.6 ‰. Previously published ε<sub>Rubisco</sub> values for CO<sub>2</sub> fixation catalyzed by *S. oleracea* Rubisco range from 28.2 to 30.3‰ (Roeske and O'Leary, 1984; Guy et al., 1993; Scott et al., 2004b), and our results fall within this range, demonstrating excellent agreement with prior work. Control experiments processed identically to the reaction assays — Control 1 (without Rubisco extract) and Control 2 (without RuBP) — showed no significant changes in

Figure 4. Measured (symbols) and calculated (lines)  $\delta^{13}$ C values of DIC plotted against the remaining DIC fraction at different stages of the CO<sub>2</sub> fixation reaction catalyzed by Rubisco from *S. oleracea*, using the GasBench system (a) and the Apollo-Picarro system (b). The corresponding logarithmically linearized plots with fitted lines used to calculate  $\epsilon_{\text{Rubisco}}$  values are shown in panels (c) and (d). Violet diamonds and yellow triangles indicate data from control experiments lacking RuBP and Rubisco extract, respectively. Note that no DIC was consumed in these control assays, the x-axis values reflect sampling time rather than the remaining DIC fraction, illustrating the stability of the carbon isotope composition over the course of the reaction.

either DIC concentration or  $\delta^{13}$ C values over the duration of the experiment (Figure 2a and Figure 4a and b, violet and yellow symbols).

Although the difference in  $\epsilon_{Rubisco}$  values between the isotope data obtained using the Apollo-Picarro system and the GasBench system is not statistically significant in the experiments presented here, the enrichment factors derived from the GasBench were consistently slightly higher than those from the Apollo-Picarro system (see Table 1). This discrepancy likely stems from differences in sample processing: samples analyzed using the GasBench were taken directly from the reaction assay, whereas those measured on the Apollo-Picarro system were first diluted with 2 mL of reaction buffer (110 mM EPPS) prior to injection. Despite pre-bubbling this buffer with  $N_2$  overnight, small amounts of DIC remained. We routinely measured the DIC content in the buffer and applied blank corrections to the concentration data. However, no correction could be applied to the isotope

measurements, as the  $\delta^{13}$ C composition of the residual DIC was unknown. This DIC most likely originated from atmospheric CO<sub>2</sub> and was therefore highly  $^{13}$ C-depleted relative to the reaction assay DIC pool, which became progressively enriched in  $\delta^{13}$ C as DIC was consumed. As a result, the introduction of trace DIC from the buffer into the increasingly  $\delta^{13}$ C-enriched assay sample led to a slight reduction in the measured  $\delta^{13}$ C values, causing a systematic underestimation of  $\epsilon_{\rm Rubisco}$  values. This underscores the importance of thoroughly degassing all reaction buffers prior to use and continuously monitoring residual DIC levels throughout the experiment to ensure accurate isotope measurements.

Another potential source of discrepancy in  $\epsilon_{Rubisco}$  values between the Apollo-Picarro and GasBench systems arises from differences in reaction termination for concentration versus isotope measurements: GasBench samples are acidified instantaneously, whereas in the Apollo-Picarro acidification is delayed by approximately 3-4 minutes due to sample processing. Because DIC concentrations are measured on the Apollo-Picarro while isotopes are measured on the GasBench, this creates a temporal mismatch. We corrected for this by linear interpolation between the two nearest DIC measurements to estimate the DIC concentration at the time of isotope sampling. This correction is particularly critical during the initial phase of the reaction, when the rate of  $CO_2$  fixation is highest. Nonetheless, minor differences in sample handling and acidification dynamics may still contribute to small but systematic deviations in the  $\epsilon_{Rubisco}$  values derived from the two systems.

Our results demonstrate that the Apollo-Picarro system, which simultaneously measures DIC concentration and isotope composition, provides accurate and precise data. Isotope measurements from the GasBench system yielded essentially identical  $\epsilon_{\text{Rubisco}}$  values, confirming the reliability of the Apollo-Picarro results. While the GasBench was therefore not strictly necessary, we continued parallel analyses in subsequent experiments to provide independent validation and to serve as a backup in experiments where technical issues occurred with the Picarro. For completeness,  $\epsilon_{\text{Rubisco}}$  values from both systems are reported in Table 1.

## 3.4 Comparison of fully-purified and partially-purified Rubisco for isotope fractionation assays

While fully-purified Rubisco is ideal for *in vitro* isotope fractionation assays — ensuring that the observed isotopic fractionation arises solely from Rubisco activity — achieving such purity is technically demanding. The purification process is time-consuming, leads to substantial protein loss (see subsection 3.1), and often compromises enzymatic activity. These challenges are manageable for *S. oleracea*, which is easy to cultivate and contains high Rubisco levels in its leaves, but they pose significant obstacles for organisms with lower cellular Rubisco content (Boller et al., 2011). To address this, we applied a simplified partial purification protocol (see section 2) and conducted *in vitro* isotope fractionation assays using these extracts. In this section, we compare the results from partially-purified extracts with those obtained using fully-purified *S. oleracea* Rubisco (see previous section), in order to evaluate the reliability and limitations of using less purified enzyme preparations. The reaction dynamics of *in vitro*  $CO_2$  fixation catalyzed by semi-purified Rubisco closely matched those observed with fully-purified Rubisco, as shown in Figure 2b. Although reaction rates were slightly slower, they remained within a comparable range, as reflected in the fitted  $v_{max}$  values in Table 1. Model-derived product inhibitor accumulation rates were also marginally lower, likely reflecting the lower Rubisco concentration in the semi-purified assay, consistent with its reduced total protein content (Table 1). The decrease in DIC concentration during  $CO_2$  fixation was accompanied by substantial  $^{13}C$  enrichment

**Table 1.** Summary of total soluble protein ([Protein]) and Rubisco content ([Rubisco]), kinetic model parameters ( $v_{max}$ ,  $k_{acc}$ ), and carbon isotope enrichment factors ( $\epsilon_{Rubisco}$ ) for each reaction assay. Bold  $\epsilon_{Rubisco}$  values represent estimates compiled from all replicates using the Pitman estimator.

| Replicate Nr.    | $[Rubisco]^1$<br>$(\mu g ml^{-1})$ | [Protein] <sup>2</sup> $(\mu g ml^{-1})$ | $v_{max}$ $(\mu \mathrm{M \ min}^{-1})$ | $k_{ m acc}$ $(\mu{ m M~min}^{-1})$ | $\epsilon_{ m Rubisco}$ GasBench (% $_{ m c}$ ) | $\epsilon_{ m Rubisco}$ Apollo-Picarro (‰) |
|------------------|------------------------------------|------------------------------------------|-----------------------------------------|-------------------------------------|-------------------------------------------------|--------------------------------------------|
| Spinacea olero   | acea (Winter                       | Giant Santos)                            |                                         |                                     |                                                 |                                            |
| fully-purified l | Rubisco extrac                     | t                                        |                                         |                                     |                                                 |                                            |
| 1                | 234                                | $397 \pm 14$                             | 570.5                                   | 4.5                                 | n.d. <sup>3</sup>                               | n.d.                                       |
| 2                | 73                                 | $124\pm4$                                | 113.3                                   | 1.4                                 | $29.8 \pm 0.8$                                  | $29.3 \pm 0.9$                             |
| 3                | 80                                 | $144\pm11$                               | 154.4                                   | 1.9                                 | $30.7 \pm 0.6$                                  | $29.9 \pm 0.9$                             |
| 2 - 3            |                                    |                                          |                                         |                                     | $30.4 \pm 0.5$                                  | $29.6 \pm 0.7$                             |
| semi-purified l  | Rubisco extrac                     | t                                        |                                         |                                     |                                                 |                                            |
| 1                | n.d.                               | $113\pm12$                               | 78.7                                    | 0.9                                 | $30.9 \pm 0.5$                                  | $30.2 \pm 0.4$                             |
| 2                | n.d.                               | $138\pm14$                               | 101.9                                   | 1.1                                 | $31.1 \pm 0.8$                                  | $30.1 \pm 0.9$                             |
| 1 - 2            | n.d.                               |                                          |                                         |                                     | $30.9 \pm 0.5$                                  | $30.2 \pm 0.5$                             |
| Synechococcu     | s sp. (PCC 63                      | 601) with semi-puri                      | fied Rubisco extract                    |                                     |                                                 |                                            |
| 1                | n.d.                               | $1470\pm100$                             | 73.4                                    | 0.1                                 | $22.2 \pm 0.7$                                  | n.d.                                       |
| 2                | n.d.                               | $3782 \pm 199$                           | 160.3                                   | 0.2                                 | $23.4 \pm 0.7$                                  | n.d.                                       |
| 3                | n.d.                               | $3390 \pm 199$                           | 123.0                                   | 0.1                                 | $22.6 \pm 0.7$                                  | n.d.                                       |
| 1 - 3            |                                    |                                          |                                         |                                     | $22.7 \pm 0.4$                                  |                                            |
| Gephyrocapsa     | oceanica (RC                       | CC 1303) with sem                        | i-purified Rubisco extra                | act                                 |                                                 |                                            |
| 1                | n.d.                               | $577 \pm 33$                             | 22.1                                    | 0.4                                 | $13.5 \pm 0.3$                                  | n.d.                                       |
| 2                | n.d.                               | $227\pm11$                               | 20.5                                    | 1.2                                 | $12.9 \pm 0.7$                                  | $12.2\pm0.5$                               |
| 1 - 2            |                                    |                                          |                                         |                                     | $13.4 \pm 0.4$                                  |                                            |

<sup>&</sup>lt;sup>1</sup> active Rubisco concentration determined using the <sup>14</sup>C-CABP binding method (Aguiló-Nicolau et al., in preparation). <sup>2</sup> Total protein concentration measured using the Bradford assay (Bradford, 1976). <sup>3</sup> n.d. = not determined.

in the residual DIC pool (Figure A1), with  $\delta^{13}$ C values increasing by up to +60% at 78% substrate conversion. The resulting  $\epsilon_{\rm Rubisco}$  was  $30.6\pm0.5$  %, based on the average of values obtained from GasBench and Apollo-Picarro measurements. This is statistically indistinguishable from the value derived using fully-purified Rubisco ( $\epsilon_{\rm Rubisco} = 30.0\pm0.6$  %). A summary of

Figure 5. Log-log plot of the carbon isotope ratio ( $\ln R_{DIC}$ ) versus the natural logarithm of the remaining DIC concentration during  $CO_2$  fixation catalyzed by Rubisco from semi-purified (light green triangles) and fully-purified (green circles) *S. oleracea*. Data represent all replicate experiments. Linear regressions indicate no significant difference in isotope fractionation between the semi-purified and fully-purified Rubisco extracts.

these results is presented in Table 1, and the comparison is visualized in Figure 5, which shows a log-log plot of  $\delta^{13}$ C versus remaining DIC concentration for all replicates from both purification methods.

These results demonstrate that Rubisco does not need to be fully-purified using a lengthy, labor-intensive protocol. Instead, accurate and precise isotopic fractionation factors ( $\epsilon_{\rm Rubisco}$ ) can be reliably obtained using a simple, rapid, and user-friendly partial purification method. However, to ensure that the observed isotope fractionation arises solely from Rubisco activity, rigorous control experiments must accompany each assay. We recommend two types of controls. First, for each newly prepared Rubisco extract and reaction setup, a control assay should be performed under identical conditions but without the addition of RuBP. This control ensures that no other enzymes in the extract are consuming DIC independently of RuBP, which could otherwise alter the  $\delta^{13}$ C of the DIC pool. Second, when freshly synthesized RuBP is used, an additional control should be conducted to rule out any influence of impurities introduced during RuBP synthesis on either DIC concentration or  $\delta^{13}$ C values. These precautions are critical for validating that the measured isotopic fractionation exclusively reflects the Rubisco-catalyzed carboxylation reaction. In our experiments with semi-purified Rubisco, control assays showed no DIC depletion (Figure 2b) and no change in  $\delta^{13}$ C over time (Figure A1).

The fact that Rubisco only requires partial purified — thereby avoiding the substantial biomass demands and activity losses associated with full purification — enabled us to apply the established protocol to determine the  $\delta^{13}$ C enrichment factor in additional species with lower cellular Rubisco content than *S. oleracea*, as demonstrated in the following section. Nevertheless, given potential differences in cellular composition and metabolite background among taxa, careful validation through

appropriate controls remains essential, and the broader applicability of the method should be verified on a species-by-species basis.

## 3.5 Application of isotope fractionation measurement to non-model Rubisco enzymes

Using the improved methodology developed here, we determined two additional  $\epsilon_{\text{Rubisco}}$  values to test its applicability to phylogenetically diverse species beyond the model plant *S. oleracea*. To this end, we selected two aquatic microorganisms: *Synechococcus* sp., representing cyanobacteria, and *G. oceanica*, a coccolithophore.

The results showed that Rubisco from G. oceanica exhibited significantly lower  $CO_2$  fixation rates, whereas Rubisco from Synechococcus sp. performed comparably to S. oleracea, as shown in Figure 2c and d and the corresponding  $v_{max}$  values in Table 1. Despite the lower reaction rates, substantially higher amounts of total soluble protein were required in both assays (Table 1), reflecting the relatively low cellular Rubisco abundance in these species compared to S. oleracea (Aguiló-Nicolau et al., in preparation; Losh et al., 2013).

In addition, our kinetic model accurately reproduced the experimentally observed DIC depletion for Rubisco from *G. oceanica* and *Synechococcus* sp., with RMS errors of  $0.06\pm0.03$  mM and  $0.05\pm0.01$  mM, respectively (blue and red line in Figure 2b and c), consistent with the results obtained for *S. oleracea*. Yet, Rubisco from these two species appeared less susceptible to product inhibition. For *G. oceanica*, the rate of inhibitory by-product accumulation was comparable to that in *S. oleracea* (see Table 1); however, the inhibitory constant applied in our model — derived from Rubisco Form ID of *Galdieria sulphuraria* — is nearly 20 times higher (Pearce, 2006). This suggests that the effective inhibitor is much weaker, likely explaining the reduced inhibition. In contrast, for *Synechococcus* sp., the observed accumulation rates of the inhibitory by-product XBP were very low (see Table 1), consistent with previous reports (Pearce, 2006), and likely account for the minimal inhibition observed in our assays. Indeed, earlier studies indicate that such by-products are generally non-inhibitory under substrate-saturated conditions in *Synechococcus* and other Rubisco forms, including Form ID (Pearce, 2006). Additional mechanisms, such as enhanced inhibitor release (Pearce, 2006), may mitigate inhibition. Although the mild inhibition detected in our *Synechococcus* sp. and *G. oceanica* assays cannot be fully explained, it may reflect a gradual decline from CO<sub>2</sub>-saturated conditions as the reactions progressed.

The reduced sensitivity to product inhibition (in addition to higher k<sub>cat</sub> values) in these Rubisco forms allowed us to perform assays with smaller amounts of extracted Rubisco compared to *S. oleracea*, while still achieving sufficient DIC depletion without substantial slowing of the reaction. However, in some cases, the reaction times extended considerably, lasting up to 12 hours (see Figure 2c).

The carbon isotope signature of DIC during the reaction assays with G. oceanica and Synechococcus sp. Rubisco is shown as a function of the remaining DIC fraction in Figure 6a and b, respectively. The corresponding logarithmically linearized plots are presented in Figure 6c and d and exhibit strong linear correlations ( $R^2 > 0.99$ ). Rubisco from G. oceanica was associated with only moderate  $^{13}$ C enrichment, with  $\delta^{13}$ C values increasing by up to +35% at 79% substrate conversion. This led to an  $\epsilon_{\text{Rubisco}}$  value of  $13.1\pm0.7$  (average of Apollo-Picarrro and GasBench values), which is significantly lower than that of S. oleracea Rubisco, but falls within a similar range as the low fractionation reported for the coccolithophore *Emiliania* 

Figure 6. Measured (symbols) and calculated (lines)  $\delta^{13}$ C values of DIC plotted against the remaining DIC fraction during CO<sub>2</sub> fixation catalyzed by Rubisco from *G. oceanica* (a) and *Synechococcus* sp. The corresponding logarithmically linearized plots with fitted lines used to calculate  $\epsilon_{\text{Rubisco}}$  values are shown in panels (c) and (d). Violet diamonds and yellow triangles indicate data from control experiments lacking RuBP and Rubisco extract, respectively. Note that no DIC was consumed in these control assays, the x-axis values reflect sampling time rather than the remaining DIC fraction, illustrating the stability of the carbon isotope composition over the course of the reaction.

huxleyi (11.1‰; 95% CI: 9.8–12.6‰) (Boller et al., 2011). In contrast, the assay with Synechococcus sp. Rubisco exhibited substantially higher  $^{13}$ C enrichment, resulting in an  $\epsilon_{\text{Rubisco}}$  value of 22.7±0.4. This value is several ‰ lower than that of S. oleracea Rubisco but virtually identical to previously reported values for Synechococcus sp. Rubisco (22.0‰) (Guy et al., 1993).

Due to the high protein concentrations in these reaction assays, we encountered technical challenges in measuring DIC concentration and isotope composition using the Apollo-Picarro system. Upon injection, proteins in the DIC samples are denatured in the acidification chamber. This denaturation exposes hydrophobic regions of the proteins, which act as surfactants, adsorbing at air—water interfaces, reducing surface tension, and stabilizing foam (Delahaije and Wierenga, 2022). As a result, this foam is subsequently carried into the transfer lines of the Picarro system, disrupting gas flow making accurate measurements impossible. To mitigate this issue, we introduced an additional filtration step for *G. oceanica* and *Synechococcus* sp. Rubisco

435

assays. Specifically, samples were filtered using a 50 kDa Amicon Ultra-4 centrifugal filter unit (UFC8010, Merck, USA), effectively removing a substantial portion of the proteins from the DIC solution prior to injection. To confirm that this filtration step did not alter DIC concentration or isotope composition, we performed test experiments using *S. oleracea* with and without the centrifugation step and determined the corresponding  $\epsilon_{\text{Rubisco}}$  values. No significant differences were observed (data not shown).

#### 430 3.6 Isotopic fractionation, specificity, and evolutionary constraints

Form IB Rubisco of higher plants and *Synechococcus* sp., has the largest number of determinations of  $\epsilon_{\text{Rubisco}}$  of any Rubisco form (4 taxa; Figure 7) and define a range from 22.7 to 30%; our new result from coccolithophore *G. oceanica* provides a third determination for form 1D Rubisco and underscores the significantly lower Rubisco fractionation of coccolithophores and diatoms (11.1 to 18.5%). Transition state theory of Rubisco carboxylation suggests that isotopic fractionation correlates positively with enzyme specificity ( $S_{\text{c/o}}$ ) (Tcherkez et al., 2006). Because both  $CO_2$  and  $O_2$  are relatively featureless, selectivity is thought to arise in the transition state (Tcherkez et al., 2006). Variations in the transition state affect both Rubisco's selectivity and its isotopic fractionation, because more product-like carboxylation transition states, which correspond to higher specificity, feature shorter O-C bond lengths to the C-2 atom of RuBP. These shorter bonds are higher in energy and vibrational frequency, leading to larger kinetic isotope effects (Tcherkez et al., 2006; Tcherkez and Farquhar, 2005).

Across the limited dataset where both parameters have been measured, a positive correlation between  $\epsilon_{\text{Rubisco}}$  and  $S_{\text{c/o}}$  has indeed been observed, most clearly for Form IB Rubisco's, with the few available data for Forms IA and II appearing to follow the same trend (Figure 7, Table A1). However, Forms ID, deviates from this pattern. The lower  $\epsilon_{\text{Rubisco}}$  range of diatom and coccolithophores does not correspond to lower  $S_{\text{c/o}}$  compared to Form IB, and G. oceanica — for which no published  $S_{\text{c/o}}$  data exist — exhibits such a low  $\epsilon_{\text{Rubisco}}$  that even with a correspondingly low specificity it would fall outside the broader trend defined by Form 1B or 1B and 2 (Figure 7 dashed line, Table A1). As more measurements accumulate across phylogenetically diverse Rubisco families, it is becoming increasingly clear that no universal correlation exists between isotopic fractionation and specificity. Instead, different Rubisco lineages may follow distinct evolutionary trajectories, with isotope effects shaped by lineage-specific structural and mechanistic constraints.

These observations highlight the need to expand the dataset of  $\epsilon_{\text{Rubisco}}$  values beyond the handful of land plants and model organisms studied to date. Progress in this area has long been constrained by technical and experimental challenges, but our protocol now offers a rapid, reliable, and broadly applicable means of determining fractionation factors across diverse lineages, including those that dominate the geological record. With this approach, it becomes possible to test whether family-specific correlations between  $\epsilon_{\text{Rubisco}}$  and  $S_{\text{c/o}}$  exist, and to systematically examine the role of environmental variables such as temperature, which strongly modulates  $S_{\text{c/o}}$  in higher plants (Galmés et al., 2015; Galmés et al., 2016). Finally, these advances will also clarify how variation in  $\epsilon_{\text{Rubisco}}$  translates into lineage- and environment-specific isotopic "signatures" preserved in the sedimentary archive, providing a stronger foundation for interpreting the carbon isotope record.

Figure 7. Isotope fractionation ( $\epsilon_{Rubisco}$ ) versus  $CO_2/O_2$  selectivity ( $S_{c/o}$ ) of Rubisco for species where both parameters have been reported. A positive correlation is evident for Forms IA, IB, and II, which fall along a common trend line, suggesting that  $\epsilon_{Rubisco}$  and  $S_{c/o}$  may be linked in these groups. By contrast, Forms IC and ID deviate from this relationship, indicating that no universal correlation exists across all Rubisco lineages. Error bars represent 95% confidence intervals where available. Full symbols highlight the  $\epsilon_{Rubisco}$  values measured in this study. The coccolithophore G oceanica, for which  $S_{c/o}$  is unknown, is represented by a dashed line  $\pm$  95% confidence interval. References for  $\epsilon_{Rubisco}$  and  $S_{c/o}$  are listed in Table A1

#### 4 Conclusions

460

In summary, we show that Rubisco isotopic fractionation factors can be measured with high accuracy and precision using a simple and rapid partial purification protocol, provided that appropriate controls are included. A newly developed kinetic model, which accounts for inhibition by side products, further strengthens experimental design by guiding the choice of enzyme concentration and sampling schedule, ensuring both high  $CO_2$  fixation and reliable  $\epsilon_{Rubisco}$  estimates. The isotope measurements obtained with the GasBench system yielded essentially the same  $\epsilon_{Rubisco}$  values as the Apollo-Picarro system, confirming the reliability of the latter. Because the Apollo-Picarro simultaneously measures DIC concentration and isotope composition, it provides a simple and cost-effective means of obtaining robust isotope fractionation factors. Moreover, our method enables

- reliable determination of  $\epsilon_{\text{Rubisco}}$  across a phylogenetically broad set of species, including those with relatively low cellular Rubisco content. The resulting  $\epsilon_{\text{Rubisco}}$  values are both precise and accurate, and they align well with previously published estimates for the same or closely related taxa. Even with only three species tested, we captured a wide range of  $\epsilon_{\text{Rubisco}}$  (13.1–30‰), nearly spanning the full spectrum reported in the literature (11.1–30.0‰) (Boller et al., 2011; Guy et al., 1993). This range underscores the versatility of our approach for probing Rubisco fractionation across diverse lineages.
- Together, these advances provide a foundation for expanding  $\epsilon_{\text{Rubisco}}$  measurements across taxa, opening the way to new insights into the mechanistic basis of isotope discrimination and strengthening the use of carbon isotope records as tracers of biological and environmental change in Earth's history.

Figure A1. Measured (symbols) and calculated (lines)  $\delta^{13}$ C values of DIC plotted against the remaining DIC fraction during  $CO_2$  fixation catalyzed by Rubisco from semi-purified *S. oleracea* extract. The corresponding logarithmically linearized plots with fitted lines used to calculate  $\epsilon_{Rubisco}$  values are shown in the lower panel. Violet diamonds and yellow triangles indicate data from control experiments lacking RuBP and Rubisco extract, respectively. Note that no DIC was consumed in these control assays, the x-axis values reflect sampling time rather than the remaining DIC fraction, illustrating the stability of the carbon isotope composition over the course of the reaction.

**Table A1.** Compilation of  $\epsilon_{\text{Rubisco}}$  and  $S_{\text{c/o}}$  values for species where both parameters have been reported, along with corresponding references.  $S_{\text{c/o}}$  values from Iñiguez et al. (2020) represent averages calculated from multiple studies listed and cited therein.

| Species                        | Rubisco form | €Rubisco (‰) | $S_{ m c/o}$      | References                                                                                                                    |
|--------------------------------|--------------|--------------|-------------------|-------------------------------------------------------------------------------------------------------------------------------|
| Prochlorococcus marinus        | 1A           | 24.0         | 59.9              | Scott et al. (2007), Iñiguez et al. (2020)                                                                                    |
| Glycine max                    | 1B           | 26.8         | 92.9              | Christeller et al. (1976), Iñiguez et al. (2020)                                                                              |
| Nicotiana tabacum              | 1B           | 28.2         | 85.9              | McNevin et al. (2007), von Caemmerer et al. (2014), Iñiguez et al. (2020)                                                     |
| Spinacia oleracea              | 1B           | 30.0         | 85.7              | this study, Iñiguez et al. (2020)                                                                                             |
| Synechococcus sp.              | 1B           | 22.7         | 43.2              | this study, Iñiguez et al. (2020)                                                                                             |
| Ralstonia eutropha             | IC           | 19.0         | 75                | Thomas et al. (2019), Horken and Tabita (1999)                                                                                |
| Rhodobacter sphaeroides        | IC           | 22.4         | 60.2              | Thomas et al. (2019), Iñiguez et al. (2020)                                                                                   |
| Skeletonema costatum           | ID           | 18.5         | 72                | Boller et al. (2015), Iñiguez et al. (2020)                                                                                   |
| Emiliania huxleyi              | ID           | 11.1         | 78                | Boller et al. (2011), Iñiguez et al. (2020)                                                                                   |
| Gephyrocapsa oceanica          | ID           | 13.1         | n.d. <sup>1</sup> | this study                                                                                                                    |
| Candidatus Promineofilum breve | I'           | 16.3         | 36.1              | Wang et al. (2023a), Banda et al. (2020)                                                                                      |
| Riftia pachyptila              | II           | 19.5         | 8.6               | Robinson et al. (2003), Iñiguez et al. (2020)                                                                                 |
| Rhodospirillum rubrum          | П            | 21.4         | 11.5              | Roeske and O'Leary (1985), Guy et al. (1993),<br>McNevin et al. (2007), von Caemmerer et al. (2014),<br>Iñiguez et al. (2020) |

<sup>&</sup>lt;sup>1</sup> n.d. = not determined

480

485

Author contributions. HS and JG conceived the idea for the manuscript. RW and PA developed the experimental design, which was executed by RW, PA, and MJ. RW analyzed the data and prepared the tables and figures. RW and PA wrote the manuscript, and all authors reviewed and edited the manuscript thoroughly.

Competing interests. The authors declare that they have no conflict of interest.

Acknowledgements. This work was financially supported by an ETH Zurich Research Grant (project code: ETH-03 19-1) funded to Heather Stoll, and by the Spanish Ministry of Sciences, Innovation and Universities, the Spanish State Research Agency and the European Regional Development Funds (project UNRAVENAR, PID2023-148523NB-I00) funded to Jeroni Galmés. Pere Aguiló-Nicolau was supported by a pre-doctoral grant from the Government of the Balearic Islands (FPI-CAIB). We thank Stewart E. Bishop at ETH Zürich and Capó-Bauçà Sebastià at Universitat de les Illes Balears–INAGEA for their technical support as well as Joy Schrepfer for helping with culturing *G. oceanica*. We thank Trinidad Garcia for technical help and organization of the radioisotope facilities at the Serveis Cientifico-Tècnics (UIB). We thank OpenAI's ChatGPT (GPT-5) for assistance with language editing and phrasing. Technical instrumentation used for plant growth and Rubisco extraction and purification was supported by Platform HiTech-INAGEA (SINCO 2022/18198) funded by Conselleria d'Educació i Universitat (Govern de les Illes Balears) and FEDER 2021-2027.

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
