# Peer review of "A simplified approach for measuring Rubisco carbon isotope fractionation and the first determination in marine haptophyte *Gephyrocapsa oceanica"

_EGUsphere, 2025_

## Author Comment (AC1)

**Answer reviewer comments for manuscript**
**`[egusphere-2025-5010]`**

Comments from the reviewers are marked as "**rev {reviewer no./comment no.}**" in italicized font. The answers of the authors are indented and follow immediately after the reviewer comments. General statements made by the reviewers are not reproduced here. highlighted and  text shows changes in the manuscript. Line numbers (L) mentioned in our replies correspond to the original single-column (unrevised) version of the manuscript.

**Reviewer 1**

**Specific comments**

**rev 2/1**  *Line 122 what is PIC?*

> PIC refers to a protease inhibitor cocktail, which is defined at its first mention in Line 113.

**rev 2/2**  *Line 425 issues with sample foaming during injection into analytic systems. This is not really a comment on the manuscript, just a suggestion for subsequent analyses: try adding antifoam A to the acid instead of using centrifugal filters; there will be less chance of introducing atmospheric CO2.*

> We thank the reviewer for this helpful suggestion. We will test the addition of Antifoam A to the acid in future analyses. If successful, this approach would eliminate the centrifugation step, thereby saving time and reducing the potential for atmospheric $CO_2$ contamination.

---

## Author Comment (AC2)

**Answer reviewer comments for manuscript**
**[egusphere-2025-5010]**

Comments from the reviewers are marked as "**rev {reviewer no./comment no.}**" in italicized font. The answers of the authors are indented and follow immediately after the reviewer comments. General statements made by the reviewers are not reproduced here. highlighted and  text shows changes in the manuscript. Line numbers (L) mentioned in our replies correspond to the original single-column (unrevised) version of the manuscript.

**Reviewer 2**

**General comments**

**rev 2/1** *Your new simplified approach, consisting of semi-purified extracts and the Apollo–Picarro DIC-$\delta^{13}C$ analyzer, gives accurate results while saving time and effort. I would recommend to present this specific new methodology in the abstract. I would also recommend to introduce the approach at the end of the introduction, so that it's clear for the reader what you are going to do.*

> We agree with the reviewer and have revised the manuscript accordingly. The simplified methodological approach combining semi-purified extracts with the Apollo – Picarro $\delta^{13}$C-DIC analyzer is now explicitly described in the Abstract. In addition, we introduce this approach at the end of the Introduction to clearly outline the experimental strategy and objectives of the study for the reader. The revised part of the Abstract and Introduction are provided below.

> (L8-L10): "...Here, we present a simplified method that overcomes these limitations, eliminating the need for complex purification protocols, specialized equipment, and experimental designs that yield little $CO_2$ fixation and high uncertainties. We use a simplified purification procedure yielding semi-purified Rubisco extracts, together with an Apollo–Picarro $\delta^{13}$C-DIC analyzer capable of simultaneously measuring DIC concentration and $^{13}$C isotope ratios. Using this protocol, we accurately determined $\epsilon_{\text{Rubisco}}$ for..."

> (L80-L85): "Despite the  importance of $\epsilon_{\text{Rubisco}}$ for biogeochemical and evolutionary models, no standardized and accessible protocol exists for its determination across diverse phylogenetic groups. Here, we introduce a simplified  method that overcomes  key limitations of existing approaches and enables robust $\epsilon_{\text{Rubisco}}$ measurements across a wide range of taxa Specifically, we couple a rapid Rubisco semi-purification method to an Apollo–Picarro $\delta^{13}$C-DIC analyzer, avoiding the need for time-consuming full enzyme purification while enabling simultaneous quantification of DIC concentration and isotopic composition. We compare the performance of this semi-purified preparation with a more complex protocol yielding fully purified Rubisco, and we assess the utility of the Apollo–Picarro $\delta^{13}$C-DIC analyzer relative to classical GasBench-IRMS measurements. In addition, we incorporate a simple kinetic model to account for DIC consumption dynamics during incubations, thereby providing a rational basis for selecting appropriate enzyme

==concentrations and sampling intervals. Using this simplified approach, we demonstrate reproducibility in==  ==*Spinacia*== *oleracea* and *Synechococcus* sp. and provide  ==the first determination of== $\epsilon_{\text{Rubisco}}$ ==for the== coccolithophore *Gephyrocapsa oceanica.*"

**Specific comments**

**rev 2/2** *The methods section (2.2.) contains a description of the partial- and full purification method. However, at this point in the text it is not yet clear why these different methods are applied (e.g., why not only use the full purification method? See general comment). Only in the results section (3.1.) this becomes clear (fast/simple versus time-intensive). I would recommend integrating section 3.1 with methods section 2.2 for clarity.*

We agree that the rationale for applying both partial and full purification protocols should be clear at the Methods stage. Rather than restructuring and integrating Sections 2.2 and 3.1, we have expanded the final paragraph of the Introduction to explicitly describe the experimental strategy employed in this study (see **rev 2/1**). This revision explains the motivation for using both purification approaches – namely, the comparison of a rapid, simplified workflow with a more time-intensive full purification – and clarifies this rationale prior to the Methods section. We believe this change sufficiently addresses the reviewer's concern while preserving a clear separation between Methods and Results.

**rev 2/3** *Furthermore, section 3.1. mentions the assessment of the degree of Rubisco purification for the two methods using SDS-PAGE. This abbreviation (SDS-PAGE) is however not yet described in the text. Therefore, I would include a short explanation of SDS-PAGE in the Methods section.*

We thank the reviewer for pointing this out. To clarify the abbreviation and methodology, we have added a new subsection (Section 2.2.3) to the Methods that briefly describes SDS–PAGE and explains how it was used to assess the degree of Rubisco purification for the semi-purified and fully purified protocols.

**rev 2/4** *Lines 159 – 165 (or whole section 2.3): Can't this be integrated with section 2.4.2? It feels a bit redundant to explain the Apollo-Picarro method and GasBench method twice, in other words.*

We agree that some redundancy existed between Sections 2.3 and 2.4.2. These sections serve distinct purposes: Section 2.3 focuses on the isotope fractionation experiments and sample handling up to injection into the analytical instruments, whereas Section 2.4.2 describes the measurement procedures, calibration standards, and analytical uncertainties. We have removed redundant descriptions from both sections to avoid repetition, while keeping the methodological separation to clearly distinguish experimental execution from analytical measurement (see below).

L159-L165: "...Each sample was divided into two fractions. The first fraction, containing at least 1.25 $\mu$mol DIC (0.5-2 mL), was diluted into 2 mL of $N_2$-purged 110 mM EPPS buffer (pH 7.8) and either injected directly into a DIC-$\delta^{13}$C Analyzer  or filtered through a 50 kDa Amicon Ultra-4 centrifugal unit (UFC8010, Merck, USA) at 2,000 $\times$ g for 3 minutes at 25 °C before injection for concentration and carbon isotope analysis of DIC. The second fraction, containing at least 0.3 $\mu$mol DIC (0.1-0.5 mL), was immediately injected into a 5 mL septum-capped vial flushed with helium and preloaded with 0.1 mL of 200 mM $H_3PO_4$. These samples were analyzed the following day using a GasBench system ..."

L179-L183: "Concentration and $\delta^{13}$C composition of DIC were measured using an Apollo acidification system AS-D1 (Apollo SciTech, LLC, USA) coupled to a Picarro G2131-i cavity ring-down spectrometer (Picarro Inc., USA).  Samples were injected into the acidification chamber, where 0.9 mL of 5 M phosphoric acid was added to convert DIC into $CO_2$ gas. The evolved $CO_2$ was subsequently  sparged and transferred to the Picarro analyzer for isotopic and concentration analysis."

**rev 2/5** *Lines 214 – 215: In your case, the standard deviation of the Gaussian error propagation represents the uncertainty of the parameter ($\epsilon_{\mathrm{Rubisco}}$) estimate. Therefore, it serves as the standard error of the parameter. Perhaps it would be informative to mention this, as this makes it clearer later in the text when you're comparing calculated $\epsilon_{\mathrm{Rubisco}}$ values with other $\epsilon_{\mathrm{Rubisco}}$ values from literature. It makes statements like 'statistically indistinguishable' (line 363), 'virtually identical' (line 417), or '. . . falls within a similar range. . . ' (line 414) more substantive.*

We thank the reviewer for this important clarification. In the original manuscript, the treatment of uncertainties was described imprecisely: Gaussian error propagation was applied only to replicates measured on the same analytical instrument, whereas variability among replicates measured across both instruments was summarized using standard deviation, which we initially thought better reflected the observed between-instrument variability. We recognize that this mixed approach was confusing.

In the revised manuscript, we have standardized the uncertainty treatment by consistently applying Gaussian error propagation throughout. As a result, all reported uncertainties now represent standard errors of the $\epsilon_{\mathrm{Rubisco}}$ parameter estimates, and this clarification is explicitly stated in the Methods section (see below). Although the resulting standard errors are generally smaller than those previously reported, this change does not affect the interpretation of the data or any conclusions drawn.

In addition, where appropriate, we now apply two-tailed t-tests to statistically support comparative statements. This further substantiates statements such as "statistically indistinguishable" (see also **rev 2/8**).

L214-215: "...Measurement uncertainties were propagated using  Gaussian error propagation, and the resulting propagated standard deviations represent

the standard errors of the parameter estimates.  Comparisons of measured $\epsilon_{\text{Rubisco}}$ values were performed using two-tailed t-tests, with differences considered statistically significant at $p < 0.05$."

**rev 2/6** *Figure 2: From the figure and caption alone it is not clear that the dark and light green circles represent the two different Rubisco concentrations. This is only mentioned in line 291-292 in-text.*

We thank the reviewer for this suggestion. We have clarified the figure caption by adding the following sentence at the end of the caption for Figure 2: "...Green and dark-green circles in panel (a) represent assays performed at two different Rubisco concentrations."

**rev 2/7** *Table 1: From the table + caption alone it is not clear why the mean $\epsilon$Rubisco value of S. oleracea is only based on replicates 2 and 3. Only in lines 296 – 300 (in-text) this is clarified, as this paragraph explains which Rubisco concentrations (70 – 80 µg/ml) result in optimal experimental performance. I would recommend adding information to the table caption to clarify this.*

We thank the reviewer for this comment. To clarify the table, we have added the following sentence to the end of the table 1 caption: "...Mean $\epsilon_{\text{Rubisco}}$ value for fully purified *S. oleracea* is calculated from Replicates 2 and 3 only; Replicate 1 yielded insufficient data for a reliable estimate."

**rev 2/8** *Line 313: Please also show statistics when stating 'the difference is not statistically significant' (e.g. t-test). This also goes for line 428.*

We agree with the reviewer that statistical support should be explicitly reported when stating that differences are not statistically significant. Accordingly, we have added two-tailed t-tests to all comparisons of measured $\epsilon_{\text{Rubisco}}$ values, including those referred to in Lines 313 and 428. The corresponding statistical results are now reported in the revised manuscript.

**rev 2/9** *Considering 70% of proteins is Rubisco for the partially purified extracts, the Rubisco concentration for replicate 2 is likely around 97 µg/ml. Considering this, the vmax value is substantially lower for the partially purified Rubisco as compared to the fully purified Rubisco. What could be the reason for this? Does this mean the Rubisco in the partially purified extract is actually less catalytically active than the Rubisco from the fully purified extract? This would contradict what you state in line 261: "...approximately 63% of the Rubisco in the fully-purified extract and nearly 100% in the semi-purified extract was catalytically active". Do the impurities inhibit the activity of Rubisco? I would recommend clarifying this.*

We thank the reviewer for carefully pointing out this inconsistency. Total protein content, Rubisco concentration, and catalytic activity were assessed using three independent approaches: Bradford assay for total protein concentration, SDS–PAGE gel for Rubisco abundance, and $^{14}$CABP-binding assays for the fraction of catalytically active Rubisco.

For the fully purified Rubisco preparation, all three measurements (total protein, Rubisco content, and $^{14}$CABP-binding activity) were performed on the same extract that was subsequently used in the isotope fractionation experiments. In contrast, for the semi-purified Rubisco preparation, total protein concentration and Rubisco content were determined on the extract used for the fractionation experiments, whereas the CABP-binding assay was performed on a different semi-purified preparation obtained from an independent extraction. As a result, the $^{14}$CABP-based estimate of catalytic activity for the semi-purified extract is not directly linked to the exact preparation used for the kinetic measurements.

We therefore cannot conclusively determine whether the lower apparent $v_{max}$ of the semi-purified Rubisco reflects a reduced fraction of catalytically active enzyme, inhibitory effects of co-purifying proteins, or variability introduced by comparing different preparations. To avoid overinterpretation and speculation, we have removed the statement that "nearly 100 %" of Rubisco in the semi-purified extract was catalytically active. The revised manuscript now only reports activity estimates where measurements were performed on the same extract used for isotope fractionation (see below).

Importantly, this revision does not affect the main conclusions of the study, as the determination of $\epsilon_{Rubisco}$ is independent of $v_{max}$ values.

L260-262: "...Based on the $^{14}$C–CABP binding assay, approximately 63% of the Rubisco in the fully-purified extract  was catalytically active. For the semi-purified extract, the $^{14}$C–CABP assay indicated a high proportion of active Rubisco, but this estimate was not used quantitatively because the isotope fractionation experiment was performed on a separate extract..."

**rev 2/10** *Line 394 - 397: You implemented different values for the inhibitory constant (KI) for S. oleracea, Synechococcus, and G. oceanica, considering S. oleracea and Synechococcus are associated with Form IB, and Gephyrocapsa is associated ID. This is not entirely clear from this sentence. From first reading this sentence it seems the KI value for all species is derived from Rubisco Form ID of Galdieria sulphuraria, although this is only the case for G. oceanica. I would recommend clarifying this.*

We agree with the reviewer that the original wording was slightly ambiguous. We have revised the sentence to explicitly clarify that the inhibitory constant derived from Form ID Rubisco of *Galdieria sulphuraria* was applied only to *G. oceanica*. The revised sentence now reads as follows:

L395-L397: "...For *G. oceanica*, the rate of inhibitory by-product accumulation was comparable to that in *S. oleracea* (see Table 1); however, unlike the Form IB Rubisco of *S. oleracea* and *Synechococcus*, the inhibitory constant used for *G. oceanica* was taken from a Form ID Rubisco — specifically that of *Galdieria sulphuraria* — and  — is nearly 20 times higher..."

**Technical comment**

**rev 2/11** *Line 84: S. oleracea is not previously mentioned in the introduction using full species name. I would recommend writing it in full species name here, so Spinacia oleracea.*

We agree with the reviewer and have revised the text accordingly.

---

## Author Comment (AC3)

**Answer reviewer comments for manuscript**
`[egusphere-2025-5010]`

Comments from the reviewers are marked as "**rev {reviewer no./comment no.}**" in italicized font. The answers of the authors are indented and follow immediately after the reviewer comments. General statements made by the reviewers are not reproduced here. highlighted and  text shows changes in the manuscript. Line numbers (L) mentioned in our replies correspond to the original single-column (unrevised) version of the manuscript.

**Reviewer 3**

**High level comments**

**rev 3/1** *Regarding protocol complexity: on L8 the authors write that their simplified approach eliminates the need for "complex purification protocols, specialized equipment, and experimental designs that yield little CO2 fixation and high uncertainties." To us it seems that the cavity ringdown spectrometer is a specialized piece of equipment and its use introduces the need for additional preparatory steps (e.g., filtration, dilution) that produce some measurement artifacts (L320). It would help to simply describe what the key equipment is and why it is cheaper, simpler, or more accessible than the standard approach. This would improve the abstract, introduction (L80) and discussion.*

> We revised the abstract and the end of the Introduction to more clearly explain why the approach is cheaper, simpler, and more accessible than standard methods. These revisions make the advantages of the workflow clearer to the reader (see also **rev 2/1**).

**rev 3/2** *Moreover, as is made clear near L195, calibration of the Apollo-Picarro system was done by comparison with IRMS. If an IRMS is required for calibration, then the equipment demands of this protocol are really no simpler than the standard approach.*

> We thank the reviewer for raising this point. An IRMS is not required for determining $\epsilon_{\text{Rubisco}}$ using the Apollo–Picarro workflow. IRMS measurements were used only to independently validate the Apollo–Picarro isotope data and were not used for routine calibration or for calculating fractionation factors. We clarified this distinction by revising Abstract and end of the Introduction (see **rev 2/1**) as well as the Methods section accordingly (see below).

> L190-192: "To independently validate the Apollo-Picarro results, the carbon isotope composition of DIC was also measured using a GasBench II system (Thermo Fisher Scientific, Germany) equipped with an autosampler (CTC Analytics AG, Switzerland), coupled to a ConFlo IV interface and a Delta V Plus isotope ratio mass spectrometer (Thermo Fisher Scientific). The same in-house $NaHCO_3$ isotope standards used in the Apollo–Picarro setup were also employed in a standard bracketing procedure..."

**rev 3/3** *Regarding novelty: the issue of rubisco purity has also been previously addressed by Estep et al. 1978 Plant Physiol but see further discussion below on why the community tends to not cite this paper. The authors should cite this work and generally avoid excessive claims of novelty. The paper is an excellent resource without. Moreover, it represents the first measurement of the G. oceanica rubisco KIE, which bolsters the low KIE value from E. huxleyi and S. costatum by Boller et al. 2011 and 2015 respectively.*

We thank the reviewer for this important point. We agree that the issue of Rubisco purity has been previously addressed by Estep et al. [6], and we have now cited this work and clarified its relevance in the Introduction (see below).

L78: "  The requirement for highly purified enzymes adds complexity and time to an already demanding protocol,  yet this requirement was experimentally validated primarily in early studies and has not been systematically re-evaluated using modern analytical approaches (Estep et al.,1978)."

**rev 3/4** *Regarding correction for rubisco side reactions (L74): As far as we understand, the side reactions are not expected to affect the KIE even though they would affect the net rate of carboxylation. Moreover, it is common and appears to be defensible to monitor rubisco reactions to ≈50% completion to fit the Rayleigh curve and derive the KIE (see Guy et al. 1993).*

We appreciate the suggestion to clarify the relevance of the correction for side reactions. We propose to clarify in lines 68-74:

L72-74: "...In cases where RuBP was limiting, the assumption of full substrate consumption becomes questionable . This can be particularity problematic when DIC concentration is not measured directly and changes in reaction rate caused by inhibitor formation  from Rubisco side reactions  are not accounted for in the estimation of DIC depletion (Wang et al., 2023a, b; Pearce, 2006)..."

**rev 3/5** *On clarity: because the methods section precedes the results, it was unclear to us what the model of rubisco inactivation is for. On first reading we thought that the model of activation state was directly related to the KIE measurement. We only realized later upon re-reading that its purpose is to estimate the right amount of rubisco to assay. This could be made clearer by simply stating the purpose of this model up front, e.g. in the methods section and appropriate text. For example, the text near L290 could be moved to the beginning of S3.2*

We thank the reviewer for this suggestion. We have clarified the overall rationale for the kinetic model in the final paragraph of the Introduction (see **rev 2/1**), noting that it accounts for DIC consumption dynamics during incubations and provides a rational basis for selecting appropriate enzyme concentrations and sampling intervals.

Additionally, at the beginning of Section 3.2 (Lines 265–268), we state: "The concentration of Rubisco used in the assay was a critical parameter, as it directly

influenced the rate of $CO_2$ consumption, the extent of dissolved inorganic carbon (DIC) depletion, and the optimal timing of sample collection. To optimize both the enzyme concentration and the sampling schedule, we developed a kinetic model to simulate DIC dynamics throughout the course of the reaction..."

And in the Methods (Section 2.6, Lines 117–118), we introduce the model as follows: "To improve calculation of the needed Rubisco concentration and optimal sampling times, we modeled the concentration dynamics of DIC during the *in vitro* Rubisco catalyzed $CO_2$ fixation assay using Michaelis-Menten kinetics... "

**rev 3/6** *A technical comment on the kinetic model of rubisco inhibition: it seems that the authors fit a 2-parameter model (unknown $k_{acc}$ and $v_{max}$) from a single time-course. It seems that this would produce ambiguous fits with high uncertainty because the same trace might be compatible with either lower $v_{max}$ or higher $k_{acc}$. Some uncertainty quantification, e.g., estimating posterior parameter ranges, would be helpful here as the authors present this fitting procedure as an integral part of their method.*

We appreciate the reviewer suggestion to clarify this. We have now detailed in line 181 that our datasets contain sufficient time points to distinguish both the curvature defining $k_{acc}$ and the peak defining $v_{max}$, therefore allowing us to constrain both paramters.

L179-L281: "...The inhibitor accumulation rate ($k_{acc}$) was treated as a free parameter. Similarly, while $v_{max}$ could in principle be constrained from known $k_{cat}$ values and Rubisco concentrations, the resulting fits were unsatisfactory, so $v_{max}$ was also treated as a free fitting parameter. Our dataset contained sufficient time points to distinguish both the curvature defining $k_{acc}$ and the peak defining $v_{max}$, therefore allowing us to constrain both parameters..."

**rev 3/7** *On transparency of analysis: we did not see any links to source code for the data analysis performed. Please publish all relevant code – this is an essential component of scientific reproducibility and especially important for a methods paper.*

We agree with the reviewer that transparency and reproducibility are essential, particularly for a methods-focused manuscript. The kinetic modeling and parameter estimation were performed using Microsoft Excel, specifically employing the built-in Solver function to minimize the root mean square (RMS) error. As such, no standalone source code exists.

However, all equations, model formulations, and fitting procedures are fully described in Section 2.5 of the Methods, which allows the analysis to be readily reproduced in other computational environments (e.g., R, Python, or MATLAB). To further improve transparency, we have revised the Methods section to explicitly state that Excel Solver was used and to describe the optimization procedure in more detail (see below).

L236-238: "...The parameters $v_{max}$ and $k_{acc}$ were estimated by fitting the model to experimental data. Parameter optimization was achieved by minimizing the root mean square deviation between the modeled and measured DIC concentrations. For this purpose, we used the built-in Solver function in Microsoft Excel."

**rev 3/8** *On citation of unpublished work: the authors cite an unpublished study of their own. This citation is not essential to any of the arguments presented and could be omitted.*

We thank the reviewer for raising this point. As described by the reviewer, the cited unpublished study is not essential for any of the arguments or conclusions presented in this manuscript. We retain the citation because this study, developed in parallel and submitted simultaneously, provides further example of the use of this method for $\epsilon_{\text{Rubisco}}$ determinations across a broader range of taxa than described here. We will update this citation when this paper progresses towards publication.

**rev 3/9** *On evolutionary constraints in section 3.6: We found this discussion of evolutionary constraints on rubisco to be out of place in an otherwise excellent methodological paper. The review of prior literature is somewhat out of date, omitting key references that posit alternative mechanisms that can affect rubisco carbon KIEs (Tcherkez et al. 2013 Biochemistry; Tcherkez et al. 2013 Plant Cell Environ; Bathellier et al. 2020 PNAS; Tcherkez and Farquhar 2021 J Plant Phys). In addition, if the authors do want to rely on the Tcherkez et al. 2006, they must also measure rubisco oxygen KIEs as a key aspect of that argument is that the oxygen KIE does not vary with specificity while the carbon KIE does. We strongly recommend that the authors omit or heavily trim this section.*

We appreciate the reviewers thoughtful comments regarding the scope and framing of Section 3.6. Although this manuscript has a strong methodological focus, it also presents new $\epsilon_{\text{Rubisco}}$ data for an additional Rubisco lineage. As is standard practice in studies reporting new $\epsilon_{\text{Rubisco}}$ values (e.g., Boller et al.[2],[3], Thomas et al.[13]), we believe it is important to place these data within the broader biological and evolutionary context of previously reported values.

That said, we agree with the reviewer that the discussion can be strengthened. In particular, we recognize that there is an active debate regarding whether biochemical trade-offs or phylogenetic constraints play the dominant role in shaping Rubisco kinetics and associated $\epsilon_{\text{Rubisco}}$. Both perspectives are supported by existing evidence, and we have revised Section 3.6 to reflect this debate more explicitly and fairly.

Specifically, we now acknowledge alternative biochemical and kinetic mechanisms proposed to influence $\epsilon_{\text{Rubisco}}$ values and have expanded Section 3.6 to include discussion of Bathellier et al.[1], Tcherkez[9], Tcherkez and Farquhar[10], Tcherkez et al.[11], which extend beyond the framework of Tcherkez et al.[12]. At the same time, we now also cite studies emphasizing the role of phylogenetic constraints (Bouvier et al.[4], Bouvier and Kelly[5]), providing a more balanced treatment of the literature.

We further acknowledge the reviewer's point that a rigorous test of the hypothesis proposed by Tcherkez et al.[12] would require measurements of the oxygen kinetic isotope effect. We now clarify that such measurements are beyond the scope of the present study and identify them as an important direction for future work.

To address these points, we have revised Section 3.6 as follows:

L445-L448: "... As more measurements accumulate across phylogenetically diverse Rubisco families, it is becoming increasingly clear that no universal correlation exists between isotopic fractionation and specificity. Instead, different Rubisco lineages may follow distinct evolutionary trajectories, with isotope effects shaped by lineage-specific structural and mechanistic constraints. Recent studies that explicitly compare Rubisco kinetic properties with evolutionary origin support this interpretation, whereas other work suggests that phylogenetic effects may play a secondary role relative to biochemical constraints (Bouvier (2021); Tcherkez (2021); Bouvier (2023)). We note that additional biochemical and kinetic mechanisms affecting $\epsilon_{\text{Rubisco}}$ have been proposed (e.g., Tcherkez (2013); Tcherkez (2013); Bathellier (2020)), and a full evaluation of these hypotheses — including measurements of oxygen kinetic isotope effects — should be considered in future studies."

**rev 3/10** *Figure 1: since these data are presented quantitatively in the text, please give the quantification in a second panel, e.g. as a bar plot.*

The purpose of Figure 1 is to provide a qualitative visual assessment of Rubisco purity in the semi-purified and fully purified extracts using SDS-PAGE. Quantitative estimates of purification are already presented and discussed in detail in the first Results section. Adding a separate quantitative panel was therefore not considered necessary, as it could overemphasize an aspect that is not central to the main objectives of the study.

Moreover, including quantitative information directly alongside the SDS-PAGE image could be misleading, as the gel itself is intended to be interpreted qualitatively, while quantification was performed using complementary approaches described in the text. We believe that the current presentation provides a clear and balanced view of both the visual and quantitative aspects of Rubisco purification.

**rev 3/11** *Figure 2: the number of colors used in the figure is excessive. A legend would help a lot. Also, the dark green marks in panel (a) are not described in the figure or caption, but only in the text. In principle, Figure 3 could be part of this figure.*

We thank the reviewer for the careful reading of the figure. We have added a sentence to the caption to clarify the meaning of the dark green marks in panel (a) (see **rev 2/6**), and we have also added a legend to Figure 2.

A consistent color code is used throughout the manuscript to represent the three different organisms and their respective controls, which helps the reader follow the experimental results. All colors have been tested for color-blind accessibility, and there are no restrictions on the number of colors used in figures. Given these considerations, we chose to retain the current color scheme, which effectively conveys the experimental distinctions.

**rev 3/12** *Figure 4: a legend would help here to define what the diamonds/triangles are.*

We agree. A legend has been added to Figure 4 to clearly define the symbols (diamonds and triangles).

**rev 3/13** *Figure 6: again, why proliferate colors?*

We refer to our response to **rev 3/11**. A consistent color scheme is used throughout the manuscript to distinguish organisms and experimental conditions, which we believe improves readability and continuity across figures. The colors were chosen to clearly separate datasets and have been checked for color-blind accessibility. Given these considerations, and because the current color scheme effectively conveys the relevant distinctions, we have left the figure unchanged.

**rev 3/14** *Figure 7: it is irresponsible to report an R2 value to a manually selected subset of the data. We strongly encourage the authors to (1) omit this fit line from the figure and (2) tone down their discussion of its evolutionary implications. There is simply too little data to draw solid conclusions from. Even in the case of rubisco reaction kinetics (e.g., Flamholz Biochem 2019), where there is far more data, such conclusions are not easy to come by.*

We do not consider including the subset and its associated $R^2$ value as irresponsible. The subset selection is not arbitrary, but based on Rubisco forms, which reflect fundamental structural and evolutionary differences among enzymes from different lineages. While some forms appear to follow a global correlation, others do not. The fit is intended solely to illustrate that no universal correlation between $\epsilon_{\mathrm{Rubisco}}$ and $S_{\mathrm{c/o}}$ is observed across the currently available data.

We have also tempered the discussion in the text to avoid overinterpreting this limited dataset and to focus on highlighting variation among lineages rather than drawing firm evolutionary conclusions (see also **rev 3/9**). The figure therefore serves as a visual aid to contextualize our new measurement, not to imply definitive evolutionary trends.

**rev 3/15** *Table A1: This table should be expanded and provided in excel or CSV. It would be helpful to specify which reference provides the KIE and which the specificity. It is worth citing multiple measurements when available and useful to report additional kinetic parameters, e.g., $k_c$at and $K_m$ values, by examination of recent meta-analyses, e.g. Flamholz et al. Biochem 2019 and Iñiguez et al.[8] Iniguez et al. 2020. Please also comment in the text and caption as to whether this collection of rubisco carbon KIEs is complete.*

We appreciate the reviewer's detailed suggestions regarding Table A1

In response to the reviewer's request for improved clarity, we have revised Table A1 to separate references for $\epsilon_{\mathrm{Rubisco}}$ and $S_{\mathrm{c/o}}$ into distinct columns, making it explicit which sources provide which parameter. We have also added a statement to the table caption noting that, to the best of our knowledge, this compilation represents a complete collection of published $\epsilon_{\mathrm{Rubisco}}$ for which corresponding Sc/o values are available at the time of writing.

However, we believe that the scope of this table should remain focused. The primary purpose of Table A1 is to document the data underlying Figure 7, which relates published $\epsilon_{\mathrm{Rubisco}}$ to $S_{\mathrm{c/o}}$. Accordingly, the table is intentionally limited to the parameters required for this comparison. Additional kinetic parameters such as $k_{\mathrm{cat}}$ and $K_m$ are not discussed elsewhere in the manuscript, and including them would go beyond the scope of the present study.

Similarly, while multiple determinations may exist for some Rubiscos, Figure 7 uses averaged values where appropriate, and Table A1 reflects the data required to support this visualization rather than all individual measurements. A comprehensive meta-analysis of Rubisco kinetics, as presented in studies such as Flamholz et al.[7], Iñiguez et al.[8] is outside the intent of the present work.

Regarding data format, we prefer to provide all supplementary information within a single SI file. We do not see a clear advantage in additionally providing the table in Excel or CSV format, as all data are already fully accessible. That said, should the editor specifically request submission in an alternative format, we would of course comply.

**Specific Comments**

**rev 3/16** *L39: refer to table A1 here.*

We agree and have added a reference to Table A1 at this point in the manuscript.

**rev 3/17** *L41: this sentence is cuttable, especially as it cites an unpublished work.*

Please see **rev 3/8**

**rev 3/18** *L45: why is KIE variation important?*

KIE variation is not "important" in itself; however, characterizing this variation is crucial because it determines how $\epsilon_{\text{Rubisco}}$ values can be applied when interpreting carbon isotope records. We clarified this point in the Introduction by adding the following sentence:

L44-45: "...This wide variation has been documented in only a limited number of species, suggesting that additional values are yet to be discovered. Characterizing this variation is essential, as $\epsilon_{\text{Rubisco}}$ directly influences interpretations of carbon isotope records used to reconstruct past biological activity and environmental conditions. However,.."

**rev 3/19** *L71: explain why this method no longer requires accounting for oxygenation-derived 3PGA.*

Because we do not measure the reaction product but instead follow isotopic changes in the substrate pool, the revised text clarifies that the substrate depletion method no longer requires accounting for oxygenation-derived 3-PGA (see below).

L71-L72 "Although the substrate depletion method no longer requires accounting for oxygenation-derived 3-PGA because the measurement focuses on isotopic changes in the substrate rather than the reaction products, some applications have reported high variability in $\epsilon_{\text{Rubisco}}$ estimates within the same species."

**rev 3/20** *L75: give reference to studies that report experiments with < 30% DIC consumption*

We added the references. We also rewrote the sentence to clarify that these studies, despite exhibiting low DIC conversion, still obtained reproducible results, and to more clearly articulate the limitations associated with low DIC consumption (see text below):

L74-L77: "...Moreover, several studies report experiments ==in which DIC consumption remained below==  30 % DIC  — and in some cases even  ==below== 6 % — ==in at least some replicates, yet still yielded reproducible $\epsilon_{Rubisco}$ values. While these results suggest the assay can yield consistent outcomes under low substrate turnover, such low conversion rates inherently reduce the reliability of the linearization required for Rayleigh fractionation, increasing uncertainty and potentially compromising the accuracy of the derived $\epsilon_{Rubisco}$ values ( Wang et al.,2023a, b; Boller et al., 2011, 2025; Thomas et al., 2019).== "

**rev 3/21** *L78: cite Estep et al. 1978 Plant Physiol for prior work on testing whether rubisco purity matters. See their Table 2 for carbon KIEs from spinach prepared to different purities; they conclude, like the authors here, that "It can be seen that fractionation is independent of enzyme purity." This is a landmark study in our field, but it is unfortunately infrequently cited because the absolute KIE values are off for reasons unrelated to the important conclusions about purity and metalation state.*

We have now cited Estep et al.[6] and revised the surrounding text to clarify its relevance (see also **rev 3/3**).

**rev 3/22** *L83: "a single instrument" – specify which instrument.*

We have specified the instrument as suggested; see **rev 2/1** for details.

**rev 3/23** *L275: worth noting other other reasons why rubisco deviates from Michaelis-Menten kinetics beyond inhibitor formation. For example, the activation state can be changed, and many organisms express catalytic chaperones (rubisco activases) that catalyze the disinhibition of the enzyme complex, etc.*

We agree that other factors can cause deviations from simple Michaelis–Menten kinetics, such as changes in Rubisco activation state. In the revised manuscript, we now acknowledge these additional mechanisms (see below). Rubisco activases, however, function primarily *in vivo* and are unlikely to affect in vitro assays, so we have not included them in this context.

L279-L181: "...Similarly, while $v_{max}$ could in principle be constrained from known $k_{cat}$ values and Rubisco concentrations, the resulting fits were unsatisfactory, so $v_{max}$ was also treated as a free fitting parameter. ==We note that other factors, such as changes in Rubisco activation state, can also contribute to deviations from simple Michaelis–Menten kinetics, but these were not considered in this model.=="

**rev 3/24** *L324: why is the dilution required?*

The dilution is required to allow injection of a larger sample volume into the Apollo–Picarro system, which improves the accuracy and precision of the isotope measurements.

L122-224: "...This discrepancy likely stems from differences in sample processing: samples analyzed using the GasBench were taken directly from the reaction assay, whereas those measured on the Apollo-Picarro system were first diluted with 2 mL of reaction buffer (110 mM EPPS) prior to injection to permit a larger injection volume, thereby improving measurement accuracy and precision...."

**rev 3/25** *Table 1: what does it mean when you write "2-3"? That you pooled samples? Please clarify in place.*

Yes. The entry "2–3" indicates that the reported $\epsilon_{\text{Rubisco}}$ value is a pooled estimate derived from replicates 2 and 3. We added a footnote to clarify this further:

Footnote: Indicates which replicates were used to calculate $\epsilon_{\text{Rubisco}}$ using the Pitman estimator.

**rev 3/26** *L425: This filtration step seems like it exposes the rubisco reaction to air, which deserves more prominent mention and discussion than it is given. Please find a place to explain why this does not affect the KIE measurement much.*

We agree that exposure to air during this step warrants careful consideration. We minimized contact with air during filtration as much as possible, and to explicitly test whether this step affected the kinetic isotope effect, we performed control experiments with *S. oleracea* Rubisco both with and without the filtration step. The resulting $\epsilon_{\text{Rubisco}}$ values were statistically indistinguishable. We have now added a two-sided t-test to quantitatively demonstrate that inclusion of the filtration step does not result in a significant change in $\epsilon_{\text{Rubisco}}$. We refer the reader to **rev 2/8** for additional details. In addition, we added the following clarification to the main text:

L424-425: "...To mitigate this issue, we introduced an additional filtration step for *G. oceanica* and *Synechococcus* sp. Rubisco assays , taking care to minimize the samples exposure to air..."

**References**

[1] Bathellier, C., Yu, L.-J., Farquhar, G. D., Coote, M. L., Lorimer, G. H., and Tcherkez, G. (2020). Ribulose 1,5-bisphosphate carboxylase/oxygenase activates o¡sub¿2¡/sub¿ by electron transfer. *Proceedings of the National Academy of Sciences*, 117(39):24234–24242.

[2] Boller, A. J., Thomas, P. J., Cavanaugh, C. M., and Scott, K. M. (2011). Low stable carbon isotope fractionation by coccolithophore rubisco. *Geochimica et Cosmochimica Acta*, 75(22):7200–7207.

[3] Boller, A. J., Thomas, P. J., Cavanaugh, C. M., and Scott, K. M. (2015). Isotopic discrimination and kinetic parameters of rubisco from the marine bloom-forming diatom, *Skeletonema costatum*. *Geobiology*, 13(1):33–43.

[4] Bouvier, J. W., Emms, D. M., Rhodes, T., Bolton, J. S., Brasnett, A., Eddershaw, A., Nielsen, J. R., Unitt, A., Whitney, S. M., and Kelly, S. (2021). Rubisco adaptation is more limited by phylogenetic constraint than by catalytic trade-off. *Molecular Biology and Evolution*, 38(7):2880–2896.

[5] Bouvier, J. W. and Kelly, S. (2023). Response to tcherkez and farquhar: Rubisco adaptation is more limited by phylogenetic constraint than by catalytic trade-off. *Journal of Plant Physiology*, 287:154021.

[6] Estep, M. F., Tabita, F. R., Parker, P. L., and Van Baalen, C. (1978). Carbon isotope fractionation by ribulose-1,5-bisophosphate carboxylase from various organisms 1. *Plant Physiology*, 61(4):680–687.

[7] Flamholz, A. I., Prywes, N., Moran, U., Davidi, D., Bar-On, Y. M., Oltrogge, L. M., Alves, R., Savage, D., and Milo, R. (2019). Revisiting trade-offs between rubisco kinetic parameters. *Biochemistry*, 58(31):3365–3376.

[8] Iñiguez, C., Capó-Bauçà, S., Niinemets, Ü., Stoll, H., Aguiló-Nicolau, P., and Galmés, J. (2020). Evolutionary trends in rubisco kinetics and their co-evolution with $co_2$ concentrating mechanisms. *The Plant Journal*, 101(4):897–918.

[9] Tcherkez, G. (2013). Modelling the reaction mechanism of ribulose-1,5-bisphosphate carboxylase/oxygenase and consequences for kinetic parameters. *Plant, Cell & Environment*, 36(9):1586–1596.

[10] Tcherkez, G. and Farquhar, G. D. (2021). Rubisco catalytic adaptation is mostly driven by photosynthetic conditions – not by phylogenetic constraints. *Journal of Plant Physiology*, 267:153554.

[11] Tcherkez, G. G. B., Bathellier, C., Stuart-Williams, H., Whitney, S., Gout, E., Bligny, R., Badger, M., and Farquhar, G. D. (2013). D2o solvent isotope effects suggest uniform energy barriers in ribulose-1,5-bisphosphate carboxylase/oxygenase catalysis. *Biochemistry*, 52(5):869–877.

[12] Tcherkez, G. G. B., Farquhar, G. D., and Andrews, T. J. (2006). Despite slow catalysis and confused substrate specificity, all ribulose bisphosphate carboxylases may be nearly perfectly optimized. *Proceedings of the National Academy of Sciences*, 103(19):7246–7251.

[13] Thomas, P. J., Boller, A. J., Satagopan, S., Tabita, F. R., Cavanaugh, C. M., and Scott, K. M. (2019). Isotope discrimination by form ic rubisco from *Ralstonia eutropha* and *Rhodobacter sphaeroides*, metabolically versatile members of 'proteobacteria' from aquatic and soil habitats. *Environmental Microbiology*, 21(1):72–80.